# Smoothing the Landscape Boosts the Signal for SGD
## Optimal Sample Complexity for Learning Single Index Models

**Alex Damian**
Princeton University
ad27@princeton.edu

Eshaan Nichani
Princeton University
eshnich@princeton.edu

Rong Ge
Duke University
rongge@cs.duke.edu

Jason D. Lee
Princeton University
jasonlee@princeton.edu

## Abstract

We focus on the task of learning a single index model $\sigma(w^\star \cdot x)$ with respect to the isotropic Gaussian distribution in $d$ dimensions. Prior work has shown that the sample complexity of learning $w^\star$ is governed by the *information exponent* $k^\star$ of the link function $\sigma$, which is defined as the index of the first nonzero Hermite coefficient of $\sigma$. Ben Arous et al. [1] showed that $n \gtrsim d^{k^\star - 1}$ samples suffice for learning $w^\star$ and that this is tight for online SGD. However, the CSQ lower bound for gradient based methods only shows that $n \gtrsim d^{k^\star/2}$ samples are necessary. In this work, we close the gap between the upper and lower bounds by showing that online SGD on a smoothed loss learns $w^\star$ with $n \gtrsim d^{k^\star/2}$ samples. We also draw connections to statistical analyses of tensor PCA and to the implicit regularization effects of minibatch SGD on empirical losses.

## 1 Introduction

Gradient descent-based algorithms are popular for deriving computational and statistical guarantees for a number of high-dimensional statistical learning problems [2, 3, 1, 4–6]. Despite the fact that the empirical loss is nonconvex and in the worst case computationally intractible to optimize, for a number of statistical learning tasks gradient-based methods still converge to good solutions with polynomial runtime and sample complexity. Analyses in these settings typically study properties of the empirical loss landscape [7], and in particular the number of samples needed for the signal of the gradient arising from the population loss to overpower the noise in some uniform sense. The sample complexity for learning with gradient descent is determined by the landscape of the empirical loss.

One setting in which the empirical loss landscape showcases rich behavior is that of learning a *single-index model*. Single index models are target functions of the form $f^*(x) = \sigma(w^\star \cdot x)$, where $w^\star \in S^{d-1}$ is the unknown relevant direction and $\sigma$ is the known link function. When the covariates are drawn from the standard $d$-dimensional Gaussian distribution, the shape of the loss landscape is governed by the *information exponent* $k^\star$ of the link function $\sigma$, which characterizes the curvature of the loss landscape around the origin. Ben Arous et al. [1] show that online stochastic gradient descent on the empirical loss can recover $w^*$ with $n \gtrsim d^{\max(1,k^\star-1)}$ samples; furthermore, they present a lower bound showing that for a class of online SGD algorithms, $d^{\max(1,k^\star-1)}$ samples are indeed necessary.

However, gradient descent can be suboptimal for various statistical learning problems, as it only relies on local information in the loss landscape and is thus prone to getting stuck in local minima. For learning a single index model, the Correlational Statistical Query (CSQ) lower bound only requires

37th Conference on Neural Information Processing Systems (NeurIPS 2023).

$d^{\max(1,k^\star/2)}$ samples to recover $w^\star$ [6, 4], which is far fewer than the number of samples required by online SGD. This gap between gradient-based methods and the CSQ lower bound is also present in the Tensor PCA problem [8]; for recovering a rank 1 $k$-tensor in $d$ dimensions, both gradient descent and the power method require $d^{\max(1,k-1)}$ samples, whereas more sophisticated spectral algorithms can match the computational lower bound of $d^{\max(1,k/2)}$ samples.

In light of the lower bound from [1], it seems hopeless for a gradient-based algorithm to match the CSQ lower bound for learning single-index models. [1] considers the regime in which SGD is simply a discretization of gradient flow, in which case the poor properties of the loss landscape with insufficient samples imply a lower bound. However, recent work has shown that SGD is not just a discretization to gradient flow, but rather that it has an additional implicit regularization effect. Specifically, [9–11] show that over short periods of time, SGD converges to a quasi-stationary distribution $N(\theta, \lambda S)$ where $\theta$ is an initial reference point, $S$ is a matrix depending on the Hessian and the noise covariance and $\lambda = \frac{\eta}{B}$ measures the strength of the noise where $\eta$ is the learning rate and $B$ is the batch size. The resulting long term dynamics therefore follow the *smoothed gradient* $\widetilde{\nabla} L(\theta) = \mathbb{E}_{z \sim N(0,S)}[\nabla L(\theta + \lambda z)]$ which has the effect of regularizing the trace of the Hessian.

This implicit regularization effect of minibatch SGD has been shown to drastically improve generalization and reduce the number of samples necessary for supervised learning tasks [12–14]. However, the connection between the smoothed landscape and the resulting sample complexity is poorly understood. Towards closing this gap, we consider directly smoothing the loss landscape in order to efficiently learn single index models. Our main result, Theorem 1, shows that for $k^\star > 2$, online SGD on the smoothed loss learns $w^\star$ in $n \gtrsim d^{k^\star/2}$ samples, which matches the correlation statistical query (CSQ) lower bound. This improves over the $n \gtrsim d^{k^\star - 1}$ lower bound for online SGD on the unsmoothed loss from Ben Arous et al. [1]. Key to our analysis is the observation that smoothing the loss landscape boosts the signal-to-noise ratio in a region around the initialization, which allows the iterates to avoid the poor local minima for the unsmoothed empirical loss. Our analysis is inspired by the implicit regularization effect of minibatch SGD, along with the partial trace algorithm for Tensor PCA which achieves the optimal $d^{k/2}$ sample complexity for computationally efficient algorithms.

The outline of our paper is as follows. In Section 3 we formalize the specific statistical learning setup, define the information exponent $k^\star$, and describe our algorithm. Section 4 contains our main theorem, and Section 5 presents a heuristic derivation for how smoothing the loss landscape increases the signal-to-noise ratio. We present empirical verification in Section 6, and in Section 7 we detail connections to tensor PCA nad minibatch SGD.

## 2   Related Work

There is a rich literature on learning single index models. Kakade et al. [15] showed that gradient descent can learn single index models when the link function is Lipschitz and monotonic and designed an alternative algorithm to handle the case when the link function is unknown. Soltanolkotabi [16] focused on learning single index models where the link function is $\text{ReLU}(x) := \max(0, x)$ which has information exponent $k^\star = 1$. The phase-retrieval problem is a special case of the single index model in which the link function is $\sigma(x) = x^2$ or $\sigma(x) = |x|$; this corresponds to $k^\star = 2$, and solving phase retrieval via gradient descent has been well studied [17–20]. Dudeja and Hsu [21] constructed an algorithm which explicitly uses the harmonic structure of Hermite polynomials to identify the information exponent. Ben Arous et al. [1] provided matching upper and lower bounds that show that $n \gtrsim d^{\max(1,k^\star - 1)}$ samples are necessary and sufficient for online SGD to recover $w^\star$.

Going beyond gradient-based algorithms, Chen and Meka [22] provide an algorithm that can learn polynomials of few relevant dimensions with $n \gtrsim d$ samples, including single index models with polynomial link functions. Their estimator is based on the structure of the filtered PCA matrix $\mathbb{E}_{x,y}[\mathbf{1}_{|y| \geq \tau} x x^T]$, which relies on the heavy tails of polynomials. In particular, this upper bound does not apply to bounded link functions. Furthermore, while their result achieves the information-theoretically optimal $d$ dependence it is not a CSQ algorithm, whereas our Algorithm 1 achieves the optimal sample complexity over the class of CSQ algorithms (which contains gradient descent).

Recent work has also studied the ability of neural networks to learn single or multi-index models [5, 6, 23, 24, 4]. Bietti et al. [5] showed that two layer neural networks are able to adapt to unknown link functions with $n \gtrsim d^{k^\star}$ samples. Damian et al. [6] consider multi-index models with polynomial

link function, and under a nondegeneracy assumption which corresponds to the $k^\star = 2$ case, show that SGD on a two-layer neural network requires $n \gtrsim d^2 + r^p$ samples. Abbe et al. [24, 4] provide a generalization of the information exponent called the *leap*. They prove that in some settings, SGD can learn low dimensional target functions with $n \gtrsim d^{\text{Leap}-1}$ samples. However, they conjecture that the optimal rate is $n \gtrsim d^{\text{Leap}/2}$ and that this can be achieved by ERM rather than online SGD.

The problem of learning single index models with information exponent $k$ is strongly related to the order $k$ Tensor PCA problem (see Section 7.1), which was introduced by Richard and Montanari [8]. They conjectured the existence of a *computational-statistical gap* for Tensor PCA as the information-theoretic threshold for the problem is $n \gtrsim d$, but all known computationally efficient algorithms require $n \gtrsim d^{k/2}$. Furthermore, simple iterative estimators including tensor power method, gradient descent, and AMP are suboptimal and require $n \gtrsim d^{k-1}$ samples. Hopkins et al. [25] introduced the partial trace estimator which succeeds with $n \gtrsim d^{\lceil k/2 \rceil}$ samples. Anandkumar et al. [26] extended this result to show that gradient descent on a smoothed landscape could achieve $d^{k/2}$ sample complexity when $k = 3$ and Biroli et al. [27] heuristically extended this result to larger $k$. The success of smoothing the landscape for Tensor PCA is one of the inspirations for Algorithm 1.

## 3 Setting

### 3.1 Data distribution and target function

Our goal is to efficiently learn single index models of the form $f^\star(x) = \sigma(w^\star \cdot x)$ where $w^\star \in S^{d-1}$, the $d$-dimensional unit sphere. We assume that $\sigma$ is normalized so that $\mathbb{E}_{x \sim N(0,1)}[\sigma(x)^2] = 1$. We will also assume that $\sigma$ is differentiable and that $\sigma'$ has polynomial tails:

**Assumption 1.** *There exist constants $C_1, C_2$ such that $|\sigma'(x)| \leq C_1(1 + x^2)^{C_2}$.*

Our goal is to recover $w^\star$ given $n$ samples $(x_1, y_1), \ldots, (x_n, y_n)$ sampled i.i.d from

$$x_i \sim N(0, I_d), \quad y_i = f^\star(x_i) + z_i \quad \text{where} \quad z_i \sim N(0, \varsigma^2).$$

For simplicity of exposition, we assume that $\sigma$ is known and we take our model class to be

$$f(w, x) := \sigma(w \cdot x) \quad \text{where} \quad w \in S^{d-1}.$$

### 3.2 Algorithm: online SGD on a smoothed landscape

As $w \in S^{d-1}$ we will let $\nabla_w$ denote the spherical gradient with respect to $w$. That is, for a function $g : \mathbb{R}^d \to \mathbb{R}$, let $\nabla_w g(w) = (I - ww^T)\nabla g(z)\big|_{z=w}$ where $\nabla$ is the standard Euclidean gradient.

To compute the loss on a sample $(x, y)$, we use the correlation loss:

$$L(w; x; y) := 1 - f(w, x)y.$$

Furthermore, when the sample is omitted we refer to the population loss:

$$L(w) := \mathbb{E}_{x,y}[L(w; x; y)]$$

Our primary contribution is that SGD on a *smoothed* loss achieves the optimal sample complexity for this problem. First, we define the smoothing operator $\mathcal{L}_\lambda$:

**Definition 1.** *Let $g : S^{d-1} \to \mathbb{R}$. We define the smoothing operator $\mathcal{L}_\lambda$ by*

$$(\mathcal{L}_\lambda g)(w) := \mathbb{E}_{z \sim \mu_w}\left[g\left(\frac{w + \lambda z}{\|w + \lambda z\|}\right)\right]$$

*where $\mu_w$ is the uniform distribution over $S^{d-1}$ conditioned on being perpendicular to $w$.*

This choice of smoothing is natural for spherical gradient descent and can be directly related[1] to the Riemannian exponential map on $S^{d-1}$. We will often abuse notation and write $\mathcal{L}_\lambda(g(w))$ rather than $(\mathcal{L}_\lambda g)(w)$. The smoothed empirical loss $L_\lambda(w; x; y)$ and the population loss $L_\lambda(w)$ are defined by:

$$L_\lambda(w; x; y) := \mathcal{L}_\lambda(L(w; x; y)) \quad \text{and} \quad L_\lambda(w) := \mathcal{L}_\lambda(L(w)).$$

---

[1]This is equivalent to the intrinsic definition $(\mathcal{L}_\lambda g)(w) := \mathbb{E}_{z \sim UT_w(S^{d-1})}[\exp_w(\theta z)]$ where $\theta = \arctan(\lambda)$, $UT_w(S^{d-1})$ is the unit sphere in the tangent space $T_w(S^{d-1})$, and $\exp$ is the Riemannian exponential map.

Our algorithm is online SGD on the smoothed loss $L_\lambda$:

---

**Algorithm 1:** Smoothed Online SGD

---

**Input:** learning rate schedule $\{\eta_t\}$, smoothing schedule $\{\lambda_t\}$, steps $T$
Sample $w_0 \sim \text{Unif}(S^{d-1})$
**for** $t = 0$ *to* $T - 1$ **do**
    Sample a fresh sample $(x_t, y_t)$
    $\hat{w}_{t+1} \leftarrow w_t - \eta_t \nabla_w L_{\lambda_t}(w_t; x_t; y_t)$
    $w_{t+1} \leftarrow \hat{w}_{t+1}/\|\hat{w}_{t+1}\|$
**end**

---

### 3.3 Hermite polynomials and information exponent

The sample complexity of Algorithm 1 depends on the Hermite coefficients of $\sigma$:

**Definition 2** (Hermite Polynomials). *The $k$th Hermite polynomial $He_k : \mathbb{R} \to \mathbb{R}$ is the degree $k$, monic polynomial defined by*

$$He_k(x) = (-1)^k \frac{\nabla^k \mu(x)}{\mu(x)},$$

*where $\mu(x) := \frac{e^{-\frac{x^2}{2}}}{\sqrt{2\pi}}$ is the PDF of a standard Gaussian.*

The first few Hermite polynomials are $He_0(x) = 1, He_1(x) = x, He_2(x) = x^2 - 1, He_3(x) = x^3 - 3x$. For further discussion on the Hermite polynomials and their properties, refer to Appendix A.2. The Hermite polynomials form an orthogonal basis of $L^2(\mu)$ so any function in $L^2(\mu)$ admits a Hermite expansion. We let $\{c_k\}_{k \geq 0}$ denote the Hermite coefficients of the link function $\sigma$:

**Definition 3** (Hermite Expansion of $\sigma$). *Let $\{c_k\}_{k \geq 0}$ be the Hermite coefficients of $\sigma$, i.e.*

$$\sigma(x) = \sum_{k \geq 0} \frac{c_k}{k!} He_k(x) \quad \text{where} \quad c_k = \mathbb{E}_{x \sim N(0,1)}[\sigma(x) He_k(x)].$$

The critical quantity of interest is the *information exponent* of $\sigma$:

**Definition 4** (Information Exponent). *$k^\star = k^\star(\sigma)$ is the first index $k \geq 1$ such that $c_k \neq 0$.*

**Example 1.** *Below are some example link functions and their information exponents:*

- *$\sigma(x) = x$ and $\sigma(x) = \text{ReLU}(x) := \max(0, x)$ have information exponents $k^\star = 1$.*

- *$\sigma(x) = x^2$ and $\sigma(x) = |x|$ have information exponents $k^\star = 2$.*

- *$\sigma(x) = x^3 - 3x$ has information exponent $k^\star = 3$. More generally, $\sigma(x) = He_k(x)$ has information exponent $k^\star = k$.*

Throughout our main results we focus on the case $k^\star \geq 3$ as when $k^\star = 1, 2$, online SGD without smoothing already achieves the optimal sample complexity of $n \asymp d$ samples (up to log factors) [1].

## 4 Main Results

Our main result is a sample complexity guarantee for Algorithm 1:

**Theorem 1.** *Assume $w_0 \cdot w^\star \gtrsim d^{-1/2}$ and $\lambda \in [1, d^{1/4}]$. Then there exists a choice of $T_1, \eta$ satisfying $T_1 = \tilde{O}(d^{k^\star - 1} \lambda^{-2k^\star + 4})$ and $\eta = \tilde{O}(d^{-k^\star/2} \lambda^{2k^\star - 2})$ such that if we run Algorithm 1 with $\lambda_t = \lambda$ and $\eta_t = \eta$ for $t \leq T_1$ and $\lambda_t = 0$ and $\eta_t = O((d + t - T_1)^{-1})$ for $t > T_1$, we have that with high probability, after $T = T_1 + T_2$ steps, the final iterate $w_T$ satisfies $L(w_T) \leq O(\frac{d}{d + T_2})$.*

Note that the condition that $w_0 \cdot w^\star \gtrsim d^{1/2}$ can be guaranteed with probability $1/2$. Theorem 1 uses large smoothing (up to $\lambda = d^{1/4}$) to rapidly escape the regime in which $w \cdot w^\star \asymp d^{-1/2}$. This first stage continues until $w \cdot w^\star = 1 - o_d(1)$ which takes $T_1 = \tilde{O}(d^{k^\star/2})$ steps when $\lambda = d^{1/4}$. The second stage, in which $\lambda = 0$ and the learning rate decays linearly, lasts for an additional $T_2 = d/\epsilon$

steps where $\epsilon$ is the target accuracy. Because Algorithm 1 uses each sample exactly once, this gives the sample complexity

$$n \gtrsim d^{k^\star - 1} \lambda^{-2k^\star + 4} + d/\epsilon$$

to reach population loss $L(w_T) \leq \epsilon$. Setting $\lambda = O(1)$ is equivalent to zero smoothing and gives a sample complexity of $n \gtrsim d^{k^\star - 1} + d/\epsilon$, which matches the results of Ben Arous et al. [1]. On the other hand, setting $\lambda$ to the maximal allowable value of $d^{1/4}$ gives:

$$n \gtrsim \underbrace{d^{\frac{k^\star}{2}}}_{\substack{\text{CSQ} \\ \text{lower bound}}} + \underbrace{d/\epsilon}_{\substack{\text{information} \\ \text{lower bound}}}$$

which matches the sum of the CSQ lower bound, which is $d^{\frac{k^\star}{2}}$, and the information-theoretic lower bound, which is $d/\epsilon$, up to poly-logarithmic factors.

To complement Theorem 1, we replicate the CSQ lower bound in [6] for the specific function class $\sigma(w \cdot x)$ where $w \in S^{d-1}$. Statistical query learners are a family of learners that can query values $q(x, y)$ and receive outputs $\hat{q}$ with $|\hat{q} - \mathbb{E}_{x,y}[q(x, y)]| \leq \tau$ where $\tau$ denotes the query tolerance [28, 29]. An important class of statistical query learners is that of correlational/inner product statistical queries (CSQ) of the form $q(x, y) = yh(x)$. This includes a wide class of algorithms including gradient descent with square loss and correlation loss. For example, if the model is $f_\theta(x)$, then the gradient with square loss can be written as

$$\nabla L(w; x, y) = (f_\theta(x) - \underbrace{y)\nabla f_\theta(x)}_{\text{query}}.$$

Note that the other term in the gradient only depends on the distribution of $x \sim N(0, I_d)$ which we assume is known. However, this connection with gradient descent is only heuristic as the errors in GD are random while the errors in the SQ/CSQ framework are adversarial. However, SQ/CSQ lower bounds are frequently used to argue the existence of statistical-computational gaps in statistical learning problems [30–33]. Theorem 2 measures the tolerance needed by CSQ learners to learn $w^\star$.

**Theorem 2** (CSQ Lower Bound). *Consider the function class $\mathcal{F}_\sigma := \{\sigma(w \cdot x) : w \in S^{d-1}\}$. Any CSQ algorithm using $q$ queries requires a tolerance $\tau$ of at most*

$$\tau \lesssim \left( \frac{\log(qd)}{d} \right)^{k^\star / 4}$$

*to output an $f \in \mathcal{F}_\sigma$ with population loss less than $1/2$.*

Using the standard $\tau \approx n^{-1/2}$ heuristic which comes from concentration, this implies that $n \gtrsim d^{\frac{k^\star}{2}}$ samples are necessary to learn $\sigma(w \cdot x)$ unless the algorithm makes exponentially many queries. In the context of gradient descent, this is equivalent to either requiring exponentially many parameters or exponentially many steps of gradient descent.

## 5 Proof Sketch

In this section we highlight the key ideas of the proof of Theorem 1. The full proof is deferred to Appendix B. The proof sketch is broken into three parts. First, we conduct a general analysis on online SGD to show how the signal-to-noise ratio (SNR) affects the sample complexity. Next, we compute the SNR for the unsmoothed objective ($\lambda = 0$) to heuristically rederive the $d^{k^\star - 1}$ sample complexity in Ben Arous et al. [1]. Finally, we show how smoothing boosts the SNR and leads to an improved sample complexity of $d^{k^\star / 2}$ when $\lambda = d^{1/4}$.

### 5.1 Online SGD Analysis

To begin, we will analyze a single step of online SGD. We define $\alpha_t := w_t \cdot w^\star$ so that $\alpha_t \in [-1, 1]$ measures our current progress. Furthermore, let $v_t := -\nabla L_{\lambda_t}(w_t; x_t; y_t)$. Recall that the online SGD update is:

$$w_{t+1} = \frac{w_t + \eta_t v_t}{\|w_t + \eta_t v_t\|} \implies \alpha_{t+1} = \frac{\alpha_t + \eta_t(v_t \cdot w^\star)}{\|w_t + \eta_t v_t\|}.$$

Using the fact that $v_t \perp w_t$ and $\frac{1}{\sqrt{1+x^2}} \approx 1 - \frac{x^2}{2}$ we can Taylor expand the update for $\alpha_{t+1}$:

$$\alpha_{t+1} = \frac{\alpha_t + \eta_t(v_t \cdot w^\star)}{\sqrt{1 + \eta_t^2\|v_t\|^2}} \approx \alpha_t + \eta_t(v_t \cdot w^\star) - \frac{\eta_t^2\|v_t\|^2\alpha_t}{2} + \text{h.o.t.}$$

As in Ben Arous et al. [1], we decompose this update into a drift term and a martingale term. Let $\mathcal{F}_t = \sigma\{(x_0, y_0), \ldots, (x_{t-1}, y_{t-1})\}$ be the natural filtration. We focus on the drift term as the martingale term can be handled with standard concentration arguments. Taking expectations with respect to the fresh batch $(x_t, y_t)$ gives:

$$\mathbb{E}[\alpha_{t+1}|\mathcal{F}_t] \approx \alpha_t + \eta_t\,\mathbb{E}[v_t \cdot w^\star|\mathcal{F}_t] - \eta_t^2\,\mathbb{E}[\|v_t\|^2|\mathcal{F}_t]\alpha_t/2$$

so to guarantee a positive drift, we need to set $\eta_t \leq \frac{2\,\mathbb{E}[v_t \cdot w^\star|\mathcal{F}_t]}{\mathbb{E}[\|v_t\|^2|\mathcal{F}_t]\alpha_t}$ which gives us the value of $\eta_t$ used in Theorem 1 for $t \leq T_1$. However, to simplify the proof sketch we can assume knowledge of $\mathbb{E}[v_t \cdot w^\star|\mathcal{F}_t]$ and $\mathbb{E}[\|v_t\|^2|\mathcal{F}_t]$ and optimize over $\eta_t$ to get a maximum drift of

$$\mathbb{E}[\alpha_{t+1}|w_t] \approx \alpha_t + \frac{1}{2\alpha_t} \cdot \underbrace{\frac{\mathbb{E}[v_t \cdot w^\star|\mathcal{F}_t]^2}{\mathbb{E}[\|v_t\|^2|\mathcal{F}_t]}}_{\text{SNR}} .$$

The numerator measures the correlation of the population gradient with $w^\star$ while the denominator measures the norm of the noisy gradient. Their ratio thus has a natural interpretation as the signal-to-noise ratio (SNR). Note that the SNR is a local property, i.e. the SNR can vary for different $w_t$. When the SNR can be written as a function of $\alpha_t = w_t \cdot w^\star$, the SNR directly dictates the rate of optimization through the ODE approximation: $\alpha' \approx \text{SNR}/\alpha$. As online SGD uses each sample exactly once, the sample complexity for online SGD can be approximated by the time it takes this ODE to reach $\alpha \approx 1$ from $\alpha_0 \approx d^{-1/2}$. The remainder of the proof sketch will therefore focus on analyzing the SNR of the minibatch gradient $\nabla L_\lambda(w; x; y)$.

## 5.2 Computing the Rate with Zero Smoothing

When $\lambda = 0$, the signal and noise terms can easily be calculated. The key property we need is:

**Property 1** (Orthogonality Property). *Let $w, w^\star \in S^{d-1}$ and let $\alpha = w \cdot w^\star$. Then:*

$$\mathbb{E}_{x \sim N(0,I_d)}[He_j(w \cdot x)He_k(w^\star \cdot x)] = \delta_{jk}k!\alpha^k.$$

Using Property 1 and the Hermite expansion of $\sigma$ (Definition 3) we can directly compute the population loss and gradient. Letting $P_w^\perp := I - ww^T$ denote the projection onto the subspace orthogonal to $w$ we have:

$$L(w) = \sum_{k\geq 0} \frac{c_k^2}{k!}[1 - \alpha^k] \quad \text{and} \quad \nabla L(w) = -(P_w^\perp w^\star)\sum_{k\geq 0}\frac{c_k^2}{(k-1)!}\alpha^{k-1}.$$

As $\alpha \ll 1$ throughout most of the trajectory, the gradient is dominated by the first nonzero Hermite coefficient so up to constants: $\mathbb{E}[v \cdot w^\star] = -\nabla L(w) \cdot w^\star \approx \alpha^{k^\star - 1}$. Similarly, a standard concentration argument shows that because $v$ is a random vector in $d$ dimensions where each coordinate is $O(1)$, $\mathbb{E}[\|v\|]^2 \approx d$. Therefore the SNR is equal to $\alpha^{2(k^\star - 1)}/d$ so with an optimal learning rate schedule,

$$\mathbb{E}[\alpha_{t+1}|\mathcal{F}_t] \approx \alpha_t + \alpha_t^{2k^\star - 3}/d.$$

This can be approximated by the ODE $\alpha' = \alpha^{2k^\star - 3}/d$. Solving this ODE with the initial $\alpha_0 \asymp d^{-1/2}$ gives that the escape time is proportional to $d\alpha_0^{-2(k^\star - 1)} = d^{k^\star - 1}$ which heuristically re-derives the result of Ben Arous et al. [1].

## 5.3 How Smoothing boosts the SNR

Smoothing improves the sample complexity of online SGD by boosting the SNR of the stochastic gradient $\nabla L(w; x; y)$.

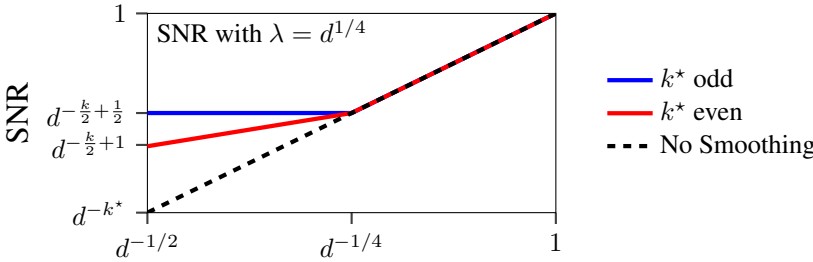

Figure 1: When $\lambda = d^{1/4}$, smoothing increases the SNR until $\alpha = \lambda d^{-1/2} = d^{-1/4}$.

**Computing the Signal** Recall that the population loss was approximately equal to $1 - \frac{c_{k^\star}^2}{k^\star!} \alpha^{k^\star}$ where $k^\star$ is the first nonzero Hermite coefficient of $\sigma$. Isolating the dominant $\alpha^{k^\star}$ term and applying the smoothing operator $\mathcal{L}_\lambda$, we get:

$$\mathcal{L}_\lambda(\alpha^{k^\star}) = \mathbb{E}_{z \sim \mu_w} \left[ \left( \frac{w + \lambda z}{\|w + \lambda z\|} \cdot w^\star \right)^{k^\star} \right].$$

Because $z \perp w$ and $\|z\| = 1$ we have that $\|w + \lambda z\| = \sqrt{1 + \lambda^2} \approx \lambda$. Therefore,

$$\mathcal{L}_\lambda(\alpha^{k^\star}) \approx \lambda^{-k^\star} \mathbb{E}_{z \sim \mu_w} \left[ (\alpha + \lambda(z \cdot w^\star))^{k^\star} \right] = \lambda^{-k^\star} \sum_{j=0}^{k} \binom{k^\star}{j} \alpha^{k^\star - j} \lambda^j \mathbb{E}_{z \sim \mu_w} [(z \cdot w^\star)^j].$$

Now because $z \overset{d}{=} -z$, the terms where $j$ is odd disappear. Furthermore, for a random $z$, $|z \cdot w^\star| = \Theta(d^{-1/2})$. Therefore reindexing and ignoring all constants we have that

$$L_\lambda(w) \approx 1 - \mathcal{L}_\lambda(\alpha^{k^\star}) \approx 1 - \lambda^{-k^\star} \sum_{j=0}^{\lfloor \frac{k^\star}{2} \rfloor} \alpha^{k^\star - 2j} \left( \lambda^2 / d \right)^j.$$

Differentiating gives that

$$\mathbb{E}[v \cdot w^\star] \approx -w^\star \cdot \nabla_w \mathcal{L}_\lambda(\alpha^{k^\star}) \approx \lambda^{-1} \sum_{j=0}^{\lfloor \frac{k^\star - 1}{2} \rfloor} \left( \frac{\alpha}{\lambda} \right)^{k^\star - 1} \left( \frac{\lambda^2}{\alpha^2 d} \right)^j.$$

As this is a geometric series, it is either dominated by the first or the last term depending on whether $\alpha \geq \lambda d^{-1/2}$ or $\alpha \leq \lambda d^{-1/2}$. Furthermore, the last term is either $d^{-\frac{k^\star - 1}{2}}$ if $k^\star$ is odd or $\frac{\alpha}{\lambda} d^{-\frac{k^\star - 2}{2}}$ if $k^\star$ is even. Therefore the signal term is:

$$\mathbb{E}[v \cdot w^\star] \approx \lambda^{-1} \begin{cases} \left( \frac{\alpha}{\lambda} \right)^{k^\star - 1} & \alpha \geq \lambda d^{-1/2} \\ d^{-\frac{k^\star - 1}{2}} & \alpha \leq \lambda d^{-1/2} \text{ and } k^\star \text{ is odd} \\ \frac{\alpha}{\lambda} d^{-\frac{k^\star - 2}{2}} & \alpha \leq \lambda d^{-1/2} \text{ and } k^\star \text{ is even} \end{cases}.$$

**Computing the Noise** Recall that $L_\lambda(w; x, y) = 1 - y\sigma(w \cdot x)$. Differentiating through the smoothing operator gives:

$$\nabla_w L_\lambda(w; x, y) = -y \nabla_w \mathcal{L}_\lambda(\sigma(w \cdot x)) \approx \lambda^{-1} yx \mathcal{L}_\lambda(\sigma'(w \cdot x)).$$

We know that with high probability, $y = \tilde{O}(1)$ and $\|x\| = O(\sqrt{d})$ so it suffices to bound $\mathcal{L}_\lambda(\sigma'(w \cdot x))$. The variance of this term is equal to:

$$\mathbb{E}_x[\mathcal{L}_\lambda(\sigma'(w \cdot x))^2] = \mathbb{E}_x \left[ \mathbb{E}_{z \sim \mu_w} \left[ \sigma' \left( \frac{w + \lambda z}{\sqrt{1 + \lambda^2}} \cdot x \right) \right]^2 \right].$$

To compute this expectation, we will create an i.i.d. copy $z'$ of $z$ and rewrite this expectation as:

$$\mathbb{E}_x[\mathcal{L}_\lambda(\sigma'(w \cdot x))^2] = \mathbb{E}_x \left[ \mathbb{E}_{z, z' \sim \mu_w} \left[ \sigma' \left( \frac{w + \lambda z}{\sqrt{1 + \lambda^2}} \cdot x \right) \sigma' \left( \frac{w + \lambda z'}{\sqrt{1 + \lambda^2}} \cdot x \right) \right] \right].$$

Now we can swap the expectations and compute the expectation with respect to $x$ first using the Hermite expansion of $\sigma$. As the first nonzero Hermite coefficient of $\sigma'$ is $k^\star - 1$, this variance is approximately equal to the correlation between $\frac{w + \lambda z}{\sqrt{1 + \lambda^2}}$ and $\frac{w + \lambda z'}{\sqrt{1 + \lambda^2}}$ raised to the $k^\star - 1$ power:

$$\mathbb{E}_x[\mathcal{L}_\lambda(\sigma'(w \cdot x))^2] \approx \mathbb{E}_{z, z' \sim \mu_w}\left[\left(\frac{w + \lambda z}{\sqrt{1 + \lambda^2}} \cdot \frac{w + \lambda z'}{\sqrt{1 + \lambda^2}}\right)^{k^\star - 1}\right] = \mathbb{E}_{z, z' \sim \mu_w}\left[\left(\frac{1 + \lambda^2 z \cdot z'}{1 + \lambda^2}\right)^{k^\star - 1}\right]$$

As $z, z'$ are random unit vectors, their inner product is of order $d^{-1/2}$. Therefore when $\lambda \ll d^{1/4}$, the first term in the numerator is dominant and when $\lambda \gg d^{1/4}$, the second term is dominant. Combining these two regimes gives that the variance is of order $\min(\lambda, d^{1/4})^{k^\star - 1}$ which motivates our choice of $\lambda = d^{1/4}$. Combining this with $y = \tilde{O}(1)$ and $\|x\| = O(\sqrt{d})$ gives that for $\lambda \le d^{1/4}$, the variance of the noise is bounded by $\mathbb{E}[\|v\|^2] \lesssim d\lambda^{-2k^\star}$. Note that in the high signal regime ($\alpha \ge \lambda d^{-1/2}$), both the signal and the noise are smaller by factors of $\lambda^{k^\star}$ which cancel when computing the SNR. However, when $\alpha \le \lambda d^{-1/2}$ the smoothing shrinks the noise faster than it shrinks the signal, resulting in an overall larger SNR. Explicitly,

$$\text{SNR} := \frac{\mathbb{E}[v \cdot w^\star]^2}{\mathbb{E}[\|v\|^2]} \approx \frac{1}{d}\begin{cases} \alpha^{2(k^\star - 1)} & \alpha \ge \lambda d^{-1/2} \\ (\lambda^2/d)^{k^\star - 1} & \alpha \le \lambda d^{-1/2} \text{ and } k^\star \text{ is odd} \\ \alpha^2(\lambda^2/d)^{k^\star - 2} & \alpha \le \lambda d^{-1/2} \text{ and } k^\star \text{ is even} \end{cases}.$$

For $\alpha \ge \lambda d^{-1/2}$, smoothing does not affect the SNR. However, when $\alpha \le \lambda d^{-1/2}$, smoothing greatly increases the SNR (see Figure 1).

Solving the ODE: $\alpha' = \text{SNR}/\alpha$ gives that it takes $T \approx d^{k^\star - 1}\lambda^{-2k^\star + 4}$ steps to converge to $\alpha \approx 1$ from $\alpha \approx d^{-1/2}$. Once $\alpha \approx 1$, the problem is locally strongly convex, so we can decay the learning rate and use classical analysis of strongly-convex functions to show that $\alpha \ge 1 - \epsilon$ with an additional $d/\epsilon$ steps, from which Theorem 1 follows.

## 6 Experiments

For $k^\star = 3, 4, 5$ and $d = 2^8, \ldots, 2^{13}$ we ran a minibatch variant of Algorithm 1 with batch size $B$ when $\sigma(x) = \frac{He_{k^\star}(x)}{\sqrt{k!}}$, the normalized $k^\star$th Hermite polynomial. We set:

$$\lambda = d^{1/4}, \quad \eta = \frac{Bd^{-k^\star/2}(1 + \lambda^2)^{k^\star - 1}}{1000k^\star!}, \quad B = \min\left(0.1d^{k^\star/2}(1 + \lambda^2)^{-2k^\star + 4}, 8192\right).$$

We computed the number of samples required for Algorithm 1 to reach $\alpha^2 = 0.5$ from $\alpha = d^{-1/2}$ and we report the min, mean, and max over 10 random seeds. For each $k$ we fit a power law of the form $n = c_1 d^{c_2}$ in order to measure how the sample complexity scales with $d$. For all values of $k^\star$, we find that $c_2 \approx k^\star/2$ which matches Theorem 1. The results can be found in Figure 2 and additional experimental details can be found in Appendix E.

## 7 Discussion

### 7.1 Tensor PCA

We next outline connections to the Tensor PCA problem. Introduced in [8], the goal of Tensor PCA is to recover the hidden direction $w^* \in \mathcal{S}^{d-1}$ from the noisy $k$-tensor $T_n \in (\mathbb{R}^d)^{\otimes k}$ given by[2]

$$T_n = (w^*)^{\otimes k} + \frac{1}{\sqrt{n}}Z,$$

where $Z \in (\mathbb{R}^d)^{\otimes k}$ is a Gaussian noise tensor with each entry drawn i.i.d from $\mathcal{N}(0, 1)$.

The Tensor PCA problem has garnered significant interest as it exhibits a *statistical-computational gap*. $w^*$ is information theoretically recoverable when $n \gtrsim d$. However, the best polynomial-time algorithms require $n \gtrsim d^{k/2}$; this lower bound has been shown to be tight for various notions of

---

[2]This normalization is equivalent to the original $\frac{1}{\beta\sqrt{d}}Z$ normalization by setting $n = \beta^2 d$.

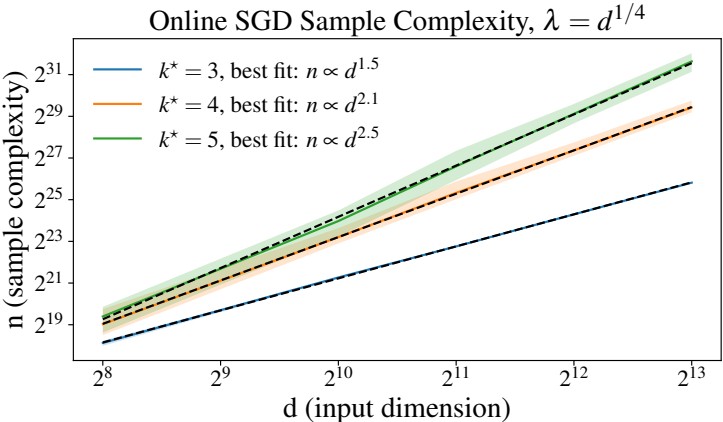

Figure 2: For $k^\star = 3, 4, 5$, Algorithm 1 finds $w^\star$ with $n \propto d^{k^\star/2}$ samples. The solid lines and the shaded areas represent the mean and min/max values over 10 random seeds. For each curve, we also fit a power law $n = c_1 d^{c_2}$ represented by the dashed lines. The value of $c_2$ is reported in the legend.

hardness such as CSQ or SoS lower bounds [34, 25, 35–39]. Tensor PCA also exhibits a gap between spectral methods and iterative algorithms. Algorithms that work in the $n \asymp d^{k/2}$ regime rely on unfolding or contracting the tensor $X$, or on semidefinite programming relaxations [34, 25]. On the other hand, iterative algorithms including gradient descent, power method, and AMP require a much larger sample complexity of $n \gtrsim d^{k-1}$ [40]. The suboptimality of iterative algorithms is believed to be due to bad properties of the landscape of the Tensor PCA objective in the region around the initialization. Specifically [41, 42] argue that there are exponentially many local minima near the equator in the $n \ll d^{k-1}$ regime. To overcome this, prior works have considered "smoothed" versions of gradient descent, and show that smoothing recovers the computationally optimal SNR in the $k = 3$ case [26] and heuristically for larger $k$ [27].

### 7.1.1 The Partial Trace Algorithm

The smoothing algorithms above are inspired by the following *partial trace* algorithm for Tensor PCA [25], which can be viewed as Algorithm 1 in the limit as $\lambda \to \infty$ [26]. Let $T_n = (w^\star)^{\otimes k} + \frac{1}{\sqrt{n}} Z$. Then we will consider iteratively contracting indices of $T$ until all that remains is a vector (if $k$ is odd) or a matrix (if $k$ is even). Explicitly, we define the partial trace tensor by

$$M_n := T_n \left( I_d^{\otimes \lceil \frac{k-2}{2} \rceil} \right) \in \begin{cases} \mathbb{R}^{d \times d} & k \text{ is even} \\ \mathbb{R}^d & k \text{ is odd}. \end{cases}$$

When $k^\star$ is odd, we can directly return $M_n$ as our estimate for $w^\star$ and when $k^\star$ is even we return the top eigenvector of $M_n$. A standard concentration argument shows that this succeeds when $n \gtrsim d^{\lceil k/2 \rceil}$. Furthermore, this can be strengthened to $d^{k/2}$ by using the partial trace vector as a warm start for gradient descent or tensor power method when $k$ is odd [26, 27].

### 7.1.2 The Connection Between Single Index Models and Tensor PCA

For both tensor PCA and learning single index models, gradient descent succeeds when the sample complexity is $n = d^{k-1}$ [40, 1]. On the other hand, the smoothing algorithms for Tensor PCA [27, 26] succeed with the computationally optimal sample complexity of $n = d^{k/2}$. Our Theorem 1 shows that this smoothing analysis can indeed be transferred to the single-index model setting.

In fact, one can make a direct connection between learning single-index models with Gaussian covariates and Tensor PCA. Consider learning a single-index model when $\sigma(x) = \frac{He_k(x)}{\sqrt{k!}}$, the normalized $k$th Hermite polynomial. Then minimizing the correlation loss is equivalent to maximizing

the loss function:

$$L_n(w) = \frac{1}{n} \sum_{i=1}^{n} y_i \frac{He_k(w \cdot x_i)}{\sqrt{k!}} = \left\langle w^{\otimes k}, T_n \right\rangle \quad \text{where} \quad T_n := \frac{1}{n} \sum_{i=1}^{n} y_i \frac{\mathbf{He}_k(x_i)}{\sqrt{k!}}.$$

Here $\mathbf{He}_k(x_i) \in (\mathbb{R}^d)^{\otimes k}$ denotes the $k$th Hermite tensor (see Appendix A.2 for background on Hermite polynomials and Hermite tensors). In addition, by the orthogonality of the Hermite tensors, $\mathbb{E}_{x,y}[T_n] = (w^\star)^{\otimes k}$ so we can decompose $T_n = (w^\star)^{\otimes k} + Z_n$ where by standard concentration, each entry of $Z_n$ is order $n^{-1/2}$. We can therefore directly apply algorithms for Tensor PCA to this problem. We remark that this connection is a heuristic, as the structure of the noise in Tensor PCA and our single index model setting are different.

## 7.2 Empirical Risk Minimization on the Smoothed Landscape

Our main sample complexity guarantee, Theorem 1, is based on a tight analysis of online SGD (Algorithm 1) in which each sample is used exactly once. One might expect that if the algorithm were allowed to reuse samples, as is standard practice in deep learning, that the algorithm could succeed with fewer samples. In particular, Abbe et al. [4] conjectured that gradient descent on the empirical loss $L_n(w) := \frac{1}{n} \sum_{i=1}^{n} L(w; x_i; y_i)$ would succeed with $n \gtrsim d^{k^\star/2}$ samples.

Our smoothing algorithm Algorithm 1 can be directly translated to the ERM setting to learn $w^\star$ with $n \gtrsim d^{k^\star/2}$ samples. We can then Taylor expand the smoothed loss in the large $\lambda$ limit:

$$\mathcal{L}_\lambda(L_n(w)) \approx \mathbb{E}_{z \sim \mathrm{Unif}(S^{d-1})} \left[ L_n(z) + \lambda^{-1} w \cdot \nabla L_n(z) + \frac{\lambda^{-2}}{2} \cdot w^T \nabla^2 L_n(z) w \right] + O(\lambda^{-3}).$$

As $\lambda \to \infty$, gradient descent on this smoothed loss will converge to $\mathbb{E}_{z \sim \mathrm{Unif}(S^{d-1})}[\nabla L_n(z)]$ which is equivalent to the partial trace estimator for $k^\star$ odd (see Section 7.1). If $k^\star$ is even, this first term is zero in expectation and gradient descent will converge to the top eigenvector of $\mathbb{E}_{z \sim \mathrm{Unif}(S^{d-1})}[\nabla^2 L_n(z)]$, which corresponds to the partial trace estimator for $k^\star$ even. Mirroring the calculation for the partial trace estimator, this succeeds with $n \gtrsim d^{\lceil k^\star/2 \rceil}$ samples. When $k^\star$ is odd, this can be further improved to $d^{k^\star/2}$ by using this estimator as a warm start from which to run gradient descent with $\lambda = 0$ as in Anandkumar et al. [26], Biroli et al. [27].

## 7.3 Connection to Minibatch SGD

A recent line of works has studied the implicit regularization effect of stochastic gradient descent [9, 11, 10]. The key idea is that over short timescales, the iterates converge to a quasi-stationary distribution $N(\theta, \lambda S)$ where $S \approx I$ depends on the Hessian and the noise covariance at $\theta$ and $\lambda$ is proportional to the ratio of the learning rate and batch size. As a result, over longer periods of time SGD follows the *smoothed gradient* of the empirical loss:

$$\widetilde{L}_n(w) = \mathbb{E}_{z \sim N(0,S)}[L_n(w + \lambda z)].$$

We therefore conjecture that minibatch SGD is also able to achieve the optimal $n \gtrsim d^{k^\star/2}$ sample complexity *without explicit smoothing* if the learning rate and batch size are properly tuned.

## Acknowledgments and Disclosure of Funding

AD acknowledges support from a NSF Graduate Research Fellowship. EN acknowledges support from a National Defense Science & Engineering Graduate Fellowship. RG is supported by NSF Award DMS-2031849, CCF-1845171 (CAREER), CCF-1934964 (Tripods) and a Sloan Research Fellowship. AD, EN, and JDL acknowledge support of the ARO under MURI Award W911NF-11-1-0304, the Sloan Research Fellowship, NSF CCF 2002272, NSF IIS 2107304, NSF CIF 2212262, ONR Young Investigator Award, and NSF CAREER Award 2144994.

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

## A  Background and Notation

### A.1  Tensor Notation

Throughout this section let $T \in (\mathbb{R}^d)^{\otimes k}$ be a $k$-tensor.

**Definition 5** (Tensor Action). *For a $j$ tensor $A \in (\mathbb{R}^d)^{\otimes j}$ with $j \leq k$, we define the action $T[A]$ of $T$ on $A$ by*

$$(T[A])_{i_1,\dots,i_{k-j}} := T_{i_1,\dots,i_k} A^{i_{k-j+1},\dots,i_k} \in (\mathbb{R}^d)^{\otimes(k-j)}.$$

We will also use $\langle T, A \rangle$ to denote $T[A] = A[T]$ when $A, T$ are both $k$ tensors. Note that this corresponds to the standard dot product after flattening $A, T$.

**Definition 6** (Permutation/Transposition). *Given a $k$-tensor $T$ and a permutation $\pi \in S_k$, we use $\pi(T)$ to denote the result of permuting the axes of $T$ by the permutation $\pi$, i.e.*

$$\pi(T)_{i_1,\ldots,i_k} := T_{i_{\pi(1)},\ldots,i_{\pi(k)}}.$$

**Definition 7** (Symmetrization). *We define $\mathrm{Sym}_k \in (\mathbb{R}^d)^{\otimes 2k}$ by*

$$(\mathrm{Sym}_k)_{i_1,\ldots,i_k,j_1,\ldots,j_k} = \frac{1}{k!} \sum_{\pi \in S_k} \delta_{i_{\pi(1)},j_1} \cdots \delta_{i_{\pi(k)},j_k}$$

*where $S_k$ is the symmetric group on $1,\ldots,k$. Note that $\mathrm{Sym}_k$ acts on $k$ tensors $T$ by*

$$(\mathrm{Sym}_k[T])_{i_1,\ldots,i_k} = \frac{1}{k!} \sum_{\pi \in S_k} \pi(T).$$

*i.e. $\mathrm{Sym}_k[T]$ is the symmetrized version of $T$.*

We will also overload notation and use $\mathrm{Sym}$ to denote the symmetrization operator, i.e. if $T$ is a $k$-tensor, $\mathrm{Sym}(T) := \mathrm{Sym}_k[T]$.

**Definition 8** (Symmetric Tensor Product). *For a $k$ tensor $T$ and a $j$ tensor $A$ we define the symmetric tensor product of $T$ and $A$ by*

$$T \widetilde{\otimes} A := \mathrm{Sym}(T \otimes A).$$

**Lemma 1.** *For any tensor $T$,*

$$\|\mathrm{Sym}(T)\|_F \le \|T\|_F.$$

*Proof.*

$$\|\mathrm{Sym}(T)\|_F = \left\| \frac{1}{k!} \sum_{\pi \in S_k} \pi(T) \right\|_F \le \frac{1}{k!} \sum_{\pi \in S_k} \|\pi(T)\|_F = \|T\|_F$$

because permuting the indices of $T$ does not change the Frobenius norm. $\square$

We will use the following two lemmas for tensor moments of the Gaussian distribution and the uniform distribution over the sphere:

**Definition 9.** *For integers $k, d > 0$, define the quantity $\nu_k^{(d)}$ as*

$$\nu_k^{(d)} := (2k-1)!! \prod_{j=0}^{k-1} \frac{1}{d+2j} = \Theta(d^{-k}).$$

*Note that $\nu_k^{(d)} = \mathbb{E}_{z \sim \mathrm{Unif}(S^{d-1})}[z_1^{2k}]$.*

**Lemma 2** (Tensorized Moments).

$$\mathbb{E}_{x \sim N(0,I_d)}[x^{\otimes 2k}] = (2k-1)!! I^{\widetilde{\otimes} k} \quad and \quad \mathbb{E}_{z \sim \mathrm{Unif}(S^{d-1})}[z^{\otimes 2k}] = \nu_k^{(d)} I^{\widetilde{\otimes} k}.$$

*Proof.* For the Gaussian moment, see [6]. The spherical moment follows from the decomposition $x \overset{d}{=} zr$ where $r \sim \chi(d)$. $\square$

**Lemma 3.**

$$\|I^{\widetilde{\otimes} j}\|_F^2 = (\nu_j^{(d)})^{-1}$$

*Proof.* By Lemma 2 we have

$$1 = \mathbb{E}_{z \sim \mathrm{Unif}(S^{d-1})}[\|z\|^{2j}] = \langle \mathbb{E}_{z \sim \mathrm{Unif}(S^{d-1})}[z^{2j}], I^{\widetilde{\otimes} j} \rangle = \nu_j^{(d)} \|I^{\widetilde{\otimes} j}\|_F^2.$$

$\square$

### A.2 Hermite Polynomials and Hermite Tensors

We provide a brief review of the properties of Hermite polynomials and Hermite tensors.

**Definition 10.** *We define the $k$th Hermite polynomial $He_k(x)$ by*

$$He_k(x) := (-1)^k \frac{\nabla^k \mu(x)}{\mu(x)}$$

*where $\mu(x) := \frac{e^{-\frac{\|x\|^2}{2}}}{(2\pi)^{d/2}}$ is the PDF of a standard Gaussian in $d$ dimensions. Note that when $d = 1$, this definition reduces to the standard univariate Hermite polynomials.*

We begin with the classical properties of the scalar Hermite polynomials:

**Lemma 4** (Properties of Hermite Polynomials). *When $d = 1$,*

- **Orthogonality:**

$$\mathbb{E}_{x \sim N(0,1)}[He_j(x)He_k(x)] = k!\delta_{jk}$$

- **Derivatives:**

$$\frac{d}{dx}He_k(x) = kHe_{k-1}(x)$$

- **Correlations:** If $x, y \sim N(0,1)$ are correlated Gaussians with correlation $\alpha := \mathbb{E}[xy]$,

$$\mathbb{E}_{x,y}[He_j(x)He_k(y)] = k!\delta_{jk}\alpha^k.$$

- **Hermite Expansion:** If $f \in L^2(\mu)$ where $\mu$ is the PDF of a standard Gaussian,

$$f(x) \overset{L^2(\mu)}{=} \sum_{k \geq 0} \frac{c_k}{k!} He_k(x) \quad \text{where} \quad c_k = \mathbb{E}_{x \sim N(0,1)}[f(x)He_k(x)].$$

These properties also have tensor analogues:

**Lemma 5** (Hermite Polynomials in Higher Dimensions).

- **Relationship to Univariante Hermite Polynomials:** *If $\|w\| = 1$,*

$$He_k(w \cdot x) = \langle He_k(x), w^{\otimes k} \rangle$$

- **Orthogonality:**

$$\mathbb{E}_{x \sim N(0,I_d)}[He_j(x) \otimes He_k(x)] = \delta_{jk}k! \operatorname{Sym}_k$$

*or equivalently, for any $j$ tensor $A$ and $k$ tensor $B$:*

$$\mathbb{E}_{x \sim N(0,I_d)}[\langle He_j(x), A \rangle \langle He_k(x), B \rangle] = \delta_{jk}k! \langle \operatorname{Sym}(A), \operatorname{Sym}(B) \rangle.$$

- **Hermite Expansions:** *If $f : \mathbb{R}^d \to \mathbb{R}$ satisfies $\mathbb{E}_{x \sim N(0,I_d)}[f(x)^2] < \infty$,*

$$f = \sum_{k \geq 0} \frac{1}{k!} \langle He_k(x), C_k \rangle \quad \text{where} \quad C_k = \mathbb{E}_{x \sim N(0,I_d)}[f(x)He_k(x)].$$

## B  Proof of Theorem 1

The proof of Theorem 1 is divided into four parts. First, Appendix B.1 introduces some notation that will be used throughout the proof. Next, Appendix B.2 computes matching upper and lower bounds for the gradient of the smoothed population loss. Similarly, Appendix B.3 concentrates the empirical gradient of the smoothed loss. Finally, Appendix B.4 combines the bounds in Appendix B.2 and Appendix B.3 with a standard online SGD analysis to arrive at the final rate.

## B.1 Additional Notation

Throughout the proof we will assume that $w \in S^{d-1}$ so that $\nabla_w$ denotes the spherical gradient of $w$. In particular, $\nabla_w g(w) \perp w$ for any $g : S^{d-1} \to \mathbb{R}$. We will also use $\alpha$ to denote $w \cdot w^\star$ so that we can write expressions such as:

$$\nabla_w \alpha = P_w^\perp w^\star = w^\star - \alpha w.$$

We will use the following assumption on $\lambda$ without reference throughout the proof:

**Assumption 2.** $\lambda^2 \leq d/C$ for a sufficiently large constant $C$.

We note that this is satisfied for the optimal choice of $\lambda = d^{1/4}$.

We will use $\tilde{O}(\cdot)$ to hide $\mathrm{polylog}(d)$ dependencies. Explicitly, $X = \tilde{O}(1)$ if there exists $C_1, C_2 > 0$ such that $|X| \leq C_1 \log(d)^{C_2}$. We will also use the following shorthand for denoting high probability events:

**Definition 11.** We say an event $E$ happens **with high probability** if for every $k \geq 0$ there exists $d(k)$ such that for all $d \geq d(k)$, $\mathbb{P}[E] \geq 1 - d^{-k}$.

Note that high probability events are closed under polynomially sized union bounds. As an example, if $X \sim N(0,1)$ then $X \leq \log(d)$ with high probability because

$$\mathbb{P}[x > \log(d)] \leq \exp\left(-\log(d)^2/2\right) \ll d^{-k}$$

for sufficiently large $d$. In general, Lemma 24 shows that if $X$ is mean zero and has polynomial tails, i.e. there exists $C$ such that $E[X^p]^{1/p} \leq p^C$, then $X = \tilde{O}(1)$ with high probability.

## B.2 Computing the Smoothed Population Gradient

Recall that

$$\sigma(x) \stackrel{L^2(\mu)}{=} \sum_{k \geq 0} \frac{c_k}{k!} He_k(x) \quad \text{where} \quad c_k := \mathbb{E}_{x \sim N(0,1)}[\sigma(x) He_k(x)].$$

In addition, because we assumed that $\mathbb{E}_{x \sim N(0,1)}[\sigma(x)^2] = 1$ we have Parseval's identity:

$$1 = \mathbb{E}_{x \sim N(0,1)}[\sigma(x)^2] = \sum_{k \geq 0} \frac{c_k^2}{k!}.$$

This Hermite decomposition immmediately implies a closed form for the population loss:

**Lemma 6** (Population Loss). Let $\alpha = w \cdot w^\star$. Then,

$$L(w) = \sum_{k \geq 0} \frac{c_k^2}{k!}[1 - \alpha^k].$$

Lemma 6 implies that to understand the smoothed population $L_\lambda(w) = (\mathcal{L}_\lambda L)(w)$, it suffices to understand $\mathcal{L}_\lambda(\alpha^k)$ for $k \geq 0$. First, we will show that the set of single index models is closed under smoothing operator $\mathcal{L}_\lambda$:

**Lemma 7.** Let $g : [-1, 1] \to \mathbb{R}$ and let $u \in S^{d-1}$. Then

$$\mathcal{L}_\lambda(g(w \cdot u)) = g_\lambda(w \cdot u)$$

where

$$g_\lambda(\alpha) := \mathbb{E}_{z \sim \mathrm{Unif}(S^{d-2})}\left[g\left(\frac{\alpha + \lambda z_1 \sqrt{1 - \alpha^2}}{\sqrt{1 + \lambda^2}}\right)\right]$$

and $z_1$ denotes the first coordinate of $z$.

*Proof.* Expanding the definition of $\mathcal{L}_\lambda$ gives:

$$\mathcal{L}_\lambda(g(w \cdot u)) = \mathbb{E}_{z \sim \mu_w}\left[ g\left( \frac{w + \lambda z}{\|w + \lambda z\|} \cdot u \right) \right]$$

$$= \mathbb{E}_{z \sim \mu_w}\left[ g\left( \frac{w \cdot u + \lambda z \cdot u}{\sqrt{1 + \lambda^2}} \right) \right].$$

Now I claim that when $z \sim \mu_w$, $z \cdot u \overset{d}{=} z_1 \sqrt{1 - (w \cdot u)^2}$ where $z \operatorname{Unif}(\sim \operatorname{Unif}(S^{d-2}))$ which would complete the proof. To see this, note that we can decompose $R^d$ into $\operatorname{span}\{w\} \oplus \operatorname{span}\{w\}^\perp$. Under this decomposition we have the polyspherical decomposition $z \overset{d}{=} (0, z')$ where $z' \sim \operatorname{Unif}(S^{d-2})$. Then

$$z \cdot u = z' \cdot P_w^\perp u \overset{d}{=} z_1 \|P_w^\perp u\| = z_1 \sqrt{1 - (w \cdot u)^2}.$$

$$\square$$

Of central interest are the quantities $\mathcal{L}_\lambda(\alpha^k)$ as these terms show up when smoothing the population loss (see Lemma 6). We begin by defining the quantity $s_k(\alpha; \lambda)$ which will provide matching upper and lower bounds on $\mathcal{L}_\lambda(\alpha^k)$ when $\alpha \geq 0$:

**Definition 12.** *We define $s_k(\alpha; \lambda)$ by*

$$s_k(\alpha; \lambda) := \frac{1}{(1 + \lambda^2)^{k/2}} \begin{cases} \alpha^k & \alpha^2 \geq \frac{\lambda^2}{d} \\ \left(\frac{\lambda^2}{d}\right)^{\frac{k}{2}} & \alpha^2 \leq \frac{\lambda^2}{d} \text{ and } k \text{ is even} \\ \alpha\left(\frac{\lambda^2}{d}\right)^{\frac{k-1}{2}} & \alpha^2 \leq \frac{\lambda^2}{d} \text{ and } k \text{ is odd} \end{cases}.$$

**Lemma 8.** *For all $k \geq 0$ and $\alpha \geq 0$, there exist constants $c(k), C(k)$ such that*

$$c(k)s_k(\alpha; \lambda) \leq \mathcal{L}_\lambda(\alpha^k) \leq C(k)s_k(\alpha; \lambda).$$

*Proof.* Using Lemma 7 we have that

$$\mathcal{L}_\lambda(\alpha^k) = \mathbb{E}_{z \sim \operatorname{Unif}(S^{d-2})}\left[ \left( \frac{\alpha + \lambda z_1 \sqrt{1 - \alpha^2}}{\sqrt{1 + \lambda^2}} \right)^k \right]$$

$$= (1 + \lambda^2)^{-k/2} \sum_{j=0}^{k} \binom{k}{j} \alpha^{k-j} \lambda^j (1 - \alpha^2)^{j/2} \mathbb{E}_{z \sim \operatorname{Unif}(S^{d-2})}[z_1^j].$$

Now note that when $j$ is odd, $\mathbb{E}_{z \sim \operatorname{Unif}(S^{d-2})}[z_1^j] = 0$ so we can re-index this sum to get

$$\mathcal{L}_\lambda(\alpha^k) = (1 + \lambda^2)^{-k/2} \sum_{j=0}^{\lfloor \frac{k}{2} \rfloor} \binom{k}{2j} \alpha^{k-2j} \lambda^{2j} (1 - \alpha^2)^j \mathbb{E}_{z \sim \operatorname{Unif}(S^{d-2})}[z_1^{2j}]$$

$$= (1 + \lambda^2)^{-k/2} \sum_{j=0}^{\lfloor \frac{k}{2} \rfloor} \binom{k}{2j} \alpha^{k-2j} \lambda^{2j} (1 - \alpha^2)^j \nu_j^{(d-1)}.$$

Note that every term in this sum is non-negative. Now we can ignore constants depending on $k$ and use that $\nu_j^{(d-1)} \asymp d^{-j}$ to get

$$\mathcal{L}_\lambda(\alpha^k) \asymp \left( \frac{\alpha}{\sqrt{1 + \lambda^2}} \right)^k \sum_{j=0}^{\lfloor \frac{k}{2} \rfloor} \left( \frac{\lambda^2(1 - \alpha^2)}{\alpha^2 d} \right)^j.$$

Now when $\alpha^2 \geq \frac{\lambda^2}{d}$, this is a decreasing geometric series which is dominated by the first term so $\mathcal{L}_\lambda(\alpha^k) \asymp \left(\frac{\alpha}{\lambda}\right)^k$. Next, when $\alpha^2 \leq \frac{\lambda^2}{d}$ we have by Assumption 2 that $\alpha \leq \frac{1}{C}$ so $1 - \alpha^2$ is bounded

away from 0. Therefore the geometric series is dominated by the last term which is

$$\frac{1}{(1+\lambda^2)^{-k/2}} \begin{cases} \left(\frac{\lambda^2}{d}\right)^{\frac{k}{2}} & k \text{ is even} \\ \alpha\left(\frac{\lambda^2}{d}\right)^{\frac{k-1}{2}} & k \text{ is odd} \end{cases}$$

which completes the proof. $\qquad\qquad\square$

Next, in order to understand the population gradient, we need to understand how the smoothing operator $\mathcal{L}_\lambda$ commutes with differentiation. We note that these do not directly commute because the smoothing distribution $\mu_w$ depends on $w$ so this term must be differentiated as well. However, smoothing and differentiation *almost* commute, which is described in the following lemma:

**Lemma 9.** *Define the dimension-dependent univariate smoothing operator by:*

$$\mathcal{L}_\lambda^{(d)} g(\alpha) := \mathbb{E}_{z\sim\mathrm{Unif}(S^{d-2})}\left[g\left(\frac{\alpha+\lambda z_1\sqrt{1-\alpha^2}}{\sqrt{1+\lambda^2}}\right)\right].$$

*Then,*

$$\frac{d}{d\alpha}\mathcal{L}_\lambda^{(d)}(g(\alpha)) = \frac{\mathcal{L}_\lambda^{(d)}(g'(\alpha))}{\sqrt{1+\lambda^2}} - \frac{\lambda^2\alpha}{(1+\lambda^2)(d-1)}\mathcal{L}_\lambda^{(d+2)}(g''(\alpha)).$$

*Proof.* Directly differentiating the definition for $\mathcal{L}_\lambda^{(d)}$ gives

$$\frac{d}{d\alpha}\mathcal{L}_\lambda^{(d)} = \frac{\mathcal{L}_\lambda^{(d)}(g'(\alpha))}{\sqrt{1+\lambda^2}} - \mathbb{E}_{z\sim\mathrm{Unif}(S^{d-2})}\left[\frac{\alpha\lambda z_1}{\sqrt{(1+\lambda^2)(1-\alpha^2)}}g'\left(\frac{\alpha+\lambda z_1\sqrt{1-\alpha^2}}{\sqrt{1+\lambda^2}}\right)\right].$$

By Lemma 25, this is equal to

$$\frac{d}{d\alpha}\mathcal{L}_\lambda^{(d)} = \frac{\mathcal{L}_\lambda^{(d)}(g'(\alpha))}{\sqrt{1+\lambda^2}} - \frac{\lambda^2\alpha}{(1+\lambda^2)(d-1)}\mathbb{E}_{z\sim\mathrm{Unif}(S^d)}\left[g''\left(\frac{\alpha+\lambda z_1\sqrt{1-\alpha^2}}{\sqrt{1+\lambda^2}}\right)\right]$$

$$= \frac{\mathcal{L}_\lambda^{(d)}(g'(\alpha))}{\sqrt{1+\lambda^2}} - \frac{\lambda^2\alpha}{(1+\lambda^2)(d-1)}\mathcal{L}_\lambda^{(d+2)}(g''(\alpha)).$$

$\qquad\qquad\square$

Now we are ready to analyze the population gradient:

**Lemma 10.**

$$\nabla_w L_\lambda(w) = -(w^\star - \alpha w)c_\lambda(\alpha)$$

*where for $\alpha \geq C^{-1/4}d^{-1/2}$,*

$$c_\lambda(\alpha) \asymp \frac{s_{k^\star-1}(\alpha;\lambda)}{\sqrt{1+\lambda^2}}.$$

*Proof.* Recall that

$$L(w) = 1 - \sum_{k\geq 0}\frac{c_k^2}{k!}\alpha^k.$$

Because $k^\star$ is the index of the first nonzero Hermite coefficient, we can start this sum at $k = k^\star$. Smoothing and differentiating gives:

$$\nabla_w L_\lambda(w) = -(w^\star - \alpha w)c_\lambda(\alpha) \quad \text{where} \quad c_\lambda(\alpha) := \sum_{k\geq k^\star}\frac{c_k^2}{k!}\frac{d}{d\alpha}\mathcal{L}_\lambda(\alpha^k).$$

We will break this into the $k = k^\star$ term and the $k > k^\star$ tail. First when $k = k^\star$ we can use Lemma 9 and Lemma 8 to get:

$$\frac{c_{k^\star}^2}{(k^\star)!} \frac{d}{d\alpha} \mathcal{L}_\lambda(\alpha^{k^\star}) = \frac{c_{k^\star}^2}{(k^\star)!} \left[ \frac{k^\star \mathcal{L}_\lambda(\alpha^{k^\star-1})}{\sqrt{1+\lambda^2}} - \frac{k^\star(k^\star-1)\lambda^2\alpha\mathcal{L}_\lambda^{(d+2)}(\alpha^{k^\star-2})}{(1+\lambda^2)(d-1)} \right].$$

The first term is equal up to constants to $\frac{s_{k^\star-1}(\alpha;\lambda)}{\sqrt{1+\lambda^2}}$ while the second term is equal up to constants to $\frac{\lambda^2\alpha}{(1+\lambda^2)d} s_{k^\star-2}(\alpha;\lambda)$. However, we have that

$$\frac{\frac{\lambda^2\alpha}{(1+\lambda^2)d} s_{k^\star-2}(\alpha;\lambda)}{\frac{s_{k^\star-1}(\alpha;\lambda)}{\sqrt{1+\lambda^2}}} = \begin{cases} \frac{\lambda^2}{d} & \alpha^2 \geq \frac{\lambda^2}{d} \\ \frac{\lambda^2}{d} & \alpha^2 \leq \frac{\lambda^2}{d} \text{ and } k^\star \text{ is even} \\ \alpha^2 & \alpha^2 \leq \frac{\lambda^2}{d} \text{ and } k^\star \text{ is odd} \end{cases} \leq \frac{\lambda^2}{d} \leq \frac{1}{C}.$$

Therefore the $k = k^\star$ term in $c_\lambda(\alpha)$ is equal up to constants to $\frac{s_{k^\star-1}(\alpha;\lambda)}{\sqrt{1+\lambda^2}}$.

Next, we handle the $k > k^\star$ tail. By Lemma 9 this is equal to

$$\sum_{k>k^\star} \frac{c_k^2}{k!} \left[ \frac{k\mathcal{L}_\lambda(\alpha^{k-1})}{\sqrt{1+\lambda^2}} - \frac{k(k-1)\lambda^2\alpha\mathcal{L}_\lambda^{(d+2)}(\alpha^{k-2})}{(1+\lambda^2)(d-1)} \right].$$

Now recall that from Lemma 8, $\mathcal{L}_\lambda(\alpha^k)$ is always non-negative so we can use $c_k^2 \leq k!$ to bound this tail in absolute value by

$$\sum_{k>k^\star} \frac{k\mathcal{L}_\lambda(\alpha^{k-1})}{\sqrt{1+\lambda^2}} + \frac{k(k-1)\lambda^2\alpha\mathcal{L}_\lambda^{(d+2)}(\alpha^{k-2})}{(1+\lambda^2)(d-1)}$$

$$\lesssim \frac{1}{\sqrt{1+\lambda^2}} \mathcal{L}_\lambda \left( \sum_{k>k^\star} k\alpha^{k-1} \right) + \frac{\lambda^2\alpha}{(1+\lambda^2)d} \mathcal{L}_\lambda^{(d+2)} \left( \sum_{k>k^\star} k(k-1)\alpha^{k-2} \right)$$

$$\lesssim \frac{1}{\sqrt{1+\lambda^2}} \mathcal{L}_\lambda \left( \frac{\alpha^{k^\star}}{(1-\alpha)^2} \right) + \frac{\lambda^2\alpha}{(1+\lambda^2)d} \mathcal{L}_\lambda^{(d+2)} \left( \frac{\alpha^{k^\star-1}}{(1-\alpha)^3} \right).$$

Now by Corollary 3, this is bounded for $d \geq 5$ by

$$\frac{s_{k^\star}(\alpha;\lambda)}{\sqrt{1+\lambda^2}} + \frac{\lambda^2\alpha}{(1+\lambda^2)d} s_{k^\star-1}(\alpha;\lambda).$$

For the first term, we have

$$\frac{s_{k^\star}(\alpha;\lambda)}{s_{k^\star-1}(\alpha;\lambda)} = \begin{cases} \frac{\alpha}{\lambda} & \alpha^2 \geq \frac{\lambda^2}{d} \\ \frac{\lambda}{d\alpha} & \alpha^2 \leq \frac{\lambda^2}{d} \text{ and } k^\star \text{ is even} \\ \frac{\alpha}{\lambda} & \alpha^2 \leq \frac{\lambda^2}{d} \text{ and } k^\star \text{ is odd} \end{cases} \leq C^{-1/4}.$$

The second term is trivially bounded by

$$\frac{\lambda^2\alpha}{(1+\lambda^2)d} \leq \frac{\lambda^2}{(1+\lambda^2)d} \leq \frac{1}{C(1+\lambda^2)} \leq \frac{1}{C\sqrt{1+\lambda^2}}$$

which completes the proof. $\qquad\square$

### B.3 Concentrating the Empirical Gradient

We cannot directly apply Lemma 7 to $\sigma(w \cdot x)$ as $\|x\| \neq 1$. Instead, we will use the properties of the Hermite tensors to directly smooth $\sigma(w \cdot x)$.

**Lemma 11.**

$$\mathcal{L}_\lambda(He_k(w \cdot x)) = \langle He_k(x), T_k(w) \rangle$$

*where*

$$T_k(w) = (1+\lambda^2)^{-\frac{k}{2}} \sum_{j=0}^{\lfloor \frac{k}{2} \rfloor} \binom{k}{2j} w^{\otimes(k-2j)} \widetilde{\otimes} (P_w^\perp)^{\widetilde{\otimes} j} \lambda^{2j} \nu_j^{(d-1)}.$$

*Proof.* Using Lemma 5, we can write

$$\mathcal{L}_\lambda(He_k(w \cdot x)) = \langle He_k(x), \mathcal{L}_\lambda(w^{\otimes k}) \rangle.$$

Now

$$\begin{aligned}
T_k(w) &= \mathcal{L}_\lambda(w^{\otimes k}) \\
&= \mathbb{E}_{z \sim \mu_w}\left[ \left( \frac{w + \lambda z}{\sqrt{1 + \lambda^2}} \right)^{\otimes k} \right] \\
&= (1 + \lambda^2)^{-\frac{k}{2}} \sum_{j=0}^{k} \binom{k}{j} w^{\otimes(k-j)} \widetilde{\otimes} \, \mathbb{E}_{z \sim \mu_w}[z^{\otimes j}] \lambda^j.
\end{aligned}$$

Now by Lemma 2, this is equal to

$$T_k(w) = (1 + \lambda^2)^{-\frac{k}{2}} \sum_{j=0}^{\lfloor \frac{k}{2} \rfloor} \binom{k}{2j} w^{\otimes(k-2j)} \widetilde{\otimes} (P_w^\perp)^{\widetilde{\otimes} j} \lambda^{2j} \nu_j^{(d-1)}$$

which completes the proof. $\qquad\square$

**Lemma 12.** *For any $u \in S^{d-1}$ with $u \perp w$,*

$$\mathbb{E}_{x \sim N(0, I_d)}\left[ (u \cdot \nabla_w \mathcal{L}_\lambda(He_k(w \cdot x)))^2 \right]$$
$$\lesssim k!\left( \frac{k^2}{1 + \lambda^2} \mathcal{L}_\lambda(\alpha^{k-1}) + \frac{\lambda^4 k^4}{(1 + \lambda^2)^2 d^2} \mathcal{L}_\lambda^{(d+2)}(\alpha^{k-2}) \right)\Bigg|_{\alpha = \frac{1}{\sqrt{1+\lambda^2}}}.$$

*Proof.* Recall that by Lemma 11 we have

$$\mathcal{L}_\lambda(He_k(w \cdot x)) = \langle He_k(x), T_k(w) \rangle$$

where

$$T_k(w) := (1 + \lambda^2)^{-\frac{k}{2}} \sum_{j=0}^{\lfloor \frac{k}{2} \rfloor} \binom{k}{2j} w^{\otimes(k-2j)} \widetilde{\otimes} (P_w^\perp)^{\widetilde{\otimes} j} \lambda^{2j} \nu_j^{(d-1)}.$$

Differentiating this with respect to $w$ gives

$$u \cdot \nabla_w \mathcal{L}_\lambda(He_k(w \cdot x)) = \langle He_k(x), \nabla_w T_k(w)[u] \rangle.$$

Now note that by Lemma 5:

$$\begin{aligned}
\mathbb{E}_{x \sim N(0, I_d)}\left[ (u \cdot \nabla_w \mathcal{L}_\lambda(He_k(w \cdot x)))^2 \right] &= \mathbb{E}_{x \sim N(0, I_d)}\left[ \langle He_k(x), \nabla_w T_k(w)[u] \rangle^2 \right] \\
&= k! \|\nabla_w T_k(w)[u]\|_F^2.
\end{aligned}$$

Therefore it suffices to compute the Frobenius norm of $\nabla_w T_k(w)[u]$. We first explicitly differentiate $T_k(w)$:

$$\begin{aligned}
\nabla_w T_k(w)[u] &= (1 + \lambda^2)^{-\frac{k}{2}} \sum_{j=0}^{\lfloor \frac{k}{2} \rfloor} \binom{k}{2j}(k - 2j) u \widetilde{\otimes} w^{\otimes k - 2j - 1} \widetilde{\otimes} (P_w^\perp)^{\widetilde{\otimes} j} \lambda^{2j} \nu_j^{(d-1)} \\
&\quad - (1 + \lambda^2)^{-\frac{k}{2}} \sum_{j=0}^{\lfloor \frac{k}{2} \rfloor} \binom{k}{2j}(2j) u \widetilde{\otimes} w^{\otimes k - 2j + 1} \widetilde{\otimes} (P_w^\perp)^{\widetilde{\otimes}(j-1)} \lambda^{2j} \nu_j^{(d-1)} \\
&= \frac{k}{(1 + \lambda^2)^{\frac{k}{2}}} \sum_{j=0}^{\lfloor \frac{k-1}{2} \rfloor} \binom{k-1}{2j} u \widetilde{\otimes} w^{\otimes k - 1 - 2j} \widetilde{\otimes} (P_w^\perp)^{\widetilde{\otimes} j} \lambda^{2j} \nu_j^{(d-1)} \\
&\quad - \frac{\lambda^2 k(k-1)}{(d-1)(1 + \lambda^2)^{\frac{k}{2}}} \sum_{j=0}^{\lfloor \frac{k-2}{2} \rfloor} \binom{k-2}{2j} u \widetilde{\otimes} w^{\otimes k - 1 - 2j} \widetilde{\otimes} (P_w^\perp)^{\widetilde{\otimes} j} \lambda^{2j} \nu_j^{(d+1)}.
\end{aligned}$$

Taking Frobenius norms gives

$$\|\nabla_w T_k(w)[u]\|_F^2$$

$$\lesssim \frac{k^2}{(1+\lambda^2)^k}\left\|\sum_{j=0}^{\lfloor\frac{k-1}{2}\rfloor}\binom{k-1}{2j}u\,\widetilde{\otimes}\,w^{\otimes k-1-2j}\,\widetilde{\otimes}(P_w^\perp)^{\widetilde{\otimes}\,j}\lambda^{2j}\nu_j^{(d-1)}\right\|_F^2$$

$$+\frac{\lambda^4 k^4}{(d-1)^2(1+\lambda^2)^k}\left\|\sum_{j=0}^{\lfloor\frac{k-2}{2}\rfloor}\binom{k-2}{2j}u\,\widetilde{\otimes}\,w^{\otimes k-1-2j}\,\widetilde{\otimes}(P_w^\perp)^{\widetilde{\otimes}\,j}\lambda^{2j}\nu_j^{(d+1)}\right\|_F^2.$$

Now we can use Lemma 1 to pull out $u$ and get:

$$\|\nabla_w T_k(w)[u]\|_F^2$$

$$\lesssim \frac{k^2}{(1+\lambda^2)^k}\left\|\sum_{j=0}^{\lfloor\frac{k-1}{2}\rfloor}\binom{k-1}{2j}w^{\otimes k-1-2j}\,\widetilde{\otimes}(P_w^\perp)^{\widetilde{\otimes}\,j}\lambda^{2j}\nu_j^{(d-1)}\right\|_F^2$$

$$+\frac{\lambda^4 k^4}{d^2(1+\lambda^2)^k}\left\|\sum_{j=0}^{\lfloor\frac{k-2}{2}\rfloor}\binom{k-2}{2j}w^{\otimes k-1-2j}\,\widetilde{\otimes}(P_w^\perp)^{\widetilde{\otimes}\,j}\lambda^{2j}\nu_j^{(d+1)}\right\|_F^2.$$

Now note that the terms in each sum are orthogonal as at least one $w$ will need to be contracted with a $P_w^\perp$. Therefore this is equivalent to:

$$\|\nabla_w T_k(w)[u]\|_F^2$$

$$\lesssim \frac{k^2}{(1+\lambda^2)^k}\sum_{j=0}^{\lfloor\frac{k-1}{2}\rfloor}\binom{k-1}{2j}^2\lambda^{4j}(\nu_j^{(d-1)})^2\left\|w^{\otimes k-1-2j}\,\widetilde{\otimes}(P_w^\perp)^{\widetilde{\otimes}\,j}\right\|_F^2$$

$$+\frac{\lambda^4 k^4}{d^2(1+\lambda^2)^k}\sum_{j=0}^{\lfloor\frac{k-2}{2}\rfloor}\binom{k-2}{2j}^2\lambda^{4j}(\nu_j^{(d+1)})^2\left\|w^{\otimes k-1-2j}\,\widetilde{\otimes}(P_w^\perp)^{\widetilde{\otimes}\,j}\right\|_F^2.$$

Next, note that for any $k$ tensor $A$, $\|\mathrm{Sym}(A)\|_F^2 = \frac{1}{k!}\sum_{\pi\in S_k}\langle A,\pi(A)\rangle$. When $A = w^{\otimes k-2j}\,\widetilde{\otimes}(P_w^\perp)^{\widetilde{\otimes}\,j}$, the only permutations that don't give $0$ are the ones which pair up all of the $w$s of which there are $(2j)!(k-2j)!$. Therefore, by Lemma 3,

$$\left\|w^{\otimes k-2j}\,\widetilde{\otimes}(P_w^\perp)^{\widetilde{\otimes}\,j}\right\|_F^2 = \frac{1}{\binom{k}{2j}}\left\|(P_w^\perp)^{\widetilde{\otimes}\,j}\right\|_F^2 = \frac{1}{\nu_j^{(d-1)}\binom{k}{2j}}.$$

Plugging this in gives:

$$\|\nabla_w T_k(w)[u]\|_F^2$$

$$\lesssim \frac{k^2}{(1+\lambda^2)^k}\sum_{j=0}^{\lfloor\frac{k-1}{2}\rfloor}\binom{k-1}{2j}\lambda^{4j}\nu_j^{(d-1)}$$

$$+\frac{\lambda^4 k^4}{d^2(1+\lambda^2)^k}\sum_{j=0}^{\lfloor\frac{k-2}{2}\rfloor}\binom{k-2}{2j}\lambda^{4j}\nu_j^{(d+1)}.$$

Now note that

$$\mathcal{L}_\lambda(\alpha^k)\Big|_{\alpha=\frac{1}{\sqrt{1+\lambda^2}}} = \frac{1}{(1+\lambda^2)^k}\sum_{k=0}^{\lfloor\frac{k}{2}\rfloor}\binom{k}{2j}\lambda^{4j}\nu_j^{(d-1)}$$

which completes the proof. $\qquad\square$

**Corollary 1.** *For any $u \in S^{d-1}$ with $u \perp w$,*

$$\mathbb{E}_{x \sim N(0, I_d)}\left[(u \cdot \nabla_w \mathcal{L}_\lambda(\sigma(w \cdot x)))^2\right] \lesssim \frac{\min\left(1 + \lambda^2, \sqrt{d}\right)^{-(k^\star - 1)}}{1 + \lambda^2}.$$

*Proof.* Note that

$$\mathcal{L}_\lambda(\sigma(w \cdot x)) = \sum_k \frac{c_k}{k!} \langle He_k(x), T_k(w) \rangle$$

so

$$u \cdot \nabla_w \mathcal{L}_\lambda(\sigma(w \cdot x)) = \sum_k \frac{c_k}{k!} \langle He_k(x), \nabla_w T_k(w)[u] \rangle.$$

By Lemma 4, these terms are orthogonal so by Lemma 12,

$$\mathbb{E}_{x \sim N(0, I_d)}\left[(u \cdot \nabla_w \mathcal{L}_\lambda(\sigma(w \cdot x)))^2\right]$$

$$\lesssim \sum_k \frac{c_k^2}{k!}\left(\frac{k^2}{1 + \lambda^2}\mathcal{L}_\lambda(\alpha^{k-1}) + \frac{\lambda^4 k^4}{(1 + \lambda^2)^2 d^2}\mathcal{L}_\lambda^{(d+2)}(\alpha^{k-2})\right)\Bigg|_{\alpha = \frac{1}{\sqrt{1+\lambda^2}}}$$

$$\lesssim \frac{1}{1 + \lambda^2}\mathcal{L}_\lambda\left(\frac{\alpha^{k^\star - 1}}{(1 - \alpha)^3}\right) + \frac{\lambda^4}{(1 + \lambda^2)^2 d^2}\mathcal{L}_\lambda^{(d+2)}\left(\frac{\alpha^{k^\star - 2}}{(1 - \alpha)^5}\right)$$

$$\lesssim \frac{1}{1 + \lambda^2}s_{k^\star - 1}\left(\frac{1}{\sqrt{1+\lambda^2}}; \lambda\right) + \frac{\lambda^4}{(1 + \lambda^2)^2 d}s_{k^\star - 2}\left(\frac{1}{\sqrt{1+\lambda^2}}; \lambda\right)$$

$$\lesssim \frac{1}{1 + \lambda^2}s_{k^\star - 1}\left(\lambda^{-1}; \lambda\right) + \frac{\lambda^4}{(1 + \lambda^2)^2 d}s_{k^\star - 2}\left(\lambda^{-1}; \lambda\right).$$

Now plugging in the formula for $s_k(\alpha; \lambda)$ gives:

$$\mathbb{E}_{x \sim N(0, I_d)}\left[(u \cdot \nabla_w \mathcal{L}_\lambda(\sigma(w \cdot x)))^2\right] \lesssim (1 + \lambda^2)^{-\frac{k^\star + 1}{2}}\begin{cases}(1 + \lambda^2)^{-\frac{k^\star - 1}{2}} & 1 + \lambda^2 \leq \sqrt{d} \\ \left(\frac{1+\lambda^2}{d}\right)^{\frac{k^\star - 1}{2}} & 1 + \lambda^2 \geq \sqrt{d}\end{cases}$$

$$\lesssim \frac{\min\left(1 + \lambda^2, \sqrt{d}\right)^{-(k^\star - 1)}}{1 + \lambda^2}.$$

$\square$

The following lemma shows that $\nabla_w \mathcal{L}_\lambda(\sigma(w \cdot x))$ inherits polynomial tails from $\sigma'$:

**Lemma 13.** *There exists an absolute constant $C$ such that for any $u \in S^{d-1}$ with $u \perp w$ and any $p \in [0, d/C]$,*

$$\mathbb{E}_{x \sim N(0, I_d)}\left[(u \cdot \nabla_w \mathcal{L}_\lambda(\sigma(w \cdot x)))^p\right]^{1/p} \lesssim \frac{p^C}{\sqrt{1 + \lambda^2}}.$$

*Proof.* Following the proof of Lemma 9 we have

$$u \cdot \nabla_w \mathcal{L}_\lambda(\sigma(w \cdot x))$$

$$= (u \cdot x)\,\mathbb{E}_{z_1 \sim \mathrm{Unif}(S^{d-2})}\left[\sigma'\left(\frac{w \cdot x + \lambda z_1 \|P_w^\perp x\|}{\sqrt{1 + \lambda^2}}\right)\left(\frac{1}{\sqrt{1 + \lambda^2}} - \frac{\lambda z_1(w \cdot x)}{\|P_w^\perp x\|\sqrt{1 + \lambda^2}}\right)\right].$$

First, we consider the first term. Its $p$ norm is bounded by

$$\frac{1}{\sqrt{1 + \lambda^2}}\,\mathbb{E}_x\left[(u \cdot x)^{2p}\right]\mathbb{E}_x\left(\mathbb{E}_{z_1 \sim \mathrm{Unif}(S^{d-2})}\left[\sigma'\left(\frac{w \cdot x + \lambda z_1 \|P_w^\perp x\|}{\sqrt{1 + \lambda^2}}\right)\right]^{2p}\right).$$

By Jensen we can pull out the expectation over $z_1$ and use Assumption 1 to get

$$\frac{1}{\sqrt{1+\lambda^2}}\,\mathbb{E}_x[(u \cdot x)^{2p}]\,\mathbb{E}_{x \sim N(0,I_d),z_1 \sim \text{Unif}(S^{d-2})}\left[\sigma'\left(\frac{w \cdot x + \lambda z_1\|P_w^\perp x\|}{\sqrt{1+\lambda^2}}\right)^{2p}\right]$$

$$= \frac{1}{\sqrt{1+\lambda^2}}\,\mathbb{E}_{x \sim N(0,1)}[x^{2p}]\,\mathbb{E}_{x \sim N(0,1)}\left[\sigma'(x)^p\right]$$

$$\lesssim \frac{\text{poly}(p)}{\sqrt{1+\lambda^2}}.$$

Similarly, the $p$ norm of the second term is bounded by

$$\frac{\lambda}{\sqrt{1+\lambda^2}} \cdot \text{poly}(p) \cdot \mathbb{E}_{x,z_1}\left[\left(\frac{z_1(x \cdot w)}{\|P_w^\perp\|}\right)^{2p}\right]^{\frac{1}{2p}} \lesssim \frac{\lambda}{d\sqrt{1+\lambda^2}}\,\text{poly}(p) \ll \frac{\text{poly}(p)}{\sqrt{1+\lambda^2}}.$$

$\square$

Finally, we can use Corollary 1 and Lemma 13 to bound the $p$ norms of the gradient:

**Lemma 14.** *Let $(x,y)$ be a fresh sample and let $v = -\nabla L_\lambda(w;x;y)$. Then there exists a constant $C$ such that for any $u \in S^{d-1}$ with $u \perp w$, any $\lambda \leq d^{1/4}$ and all $2 \leq p \leq d/C$,*

$$\mathbb{E}_{x,y}\left[(u \cdot v)^p\right]^{1/p} \lesssim \text{poly}(p) \cdot \tilde{O}\left((1+\lambda^2)^{-\frac{1}{2}-\frac{k^\star-1}{p}}\right).$$

*Proof.* First,

$$\mathbb{E}_{x,y}[(u \cdot v)^p]^{1/p} = \mathbb{E}_{x,y}[(u \cdot \nabla L_\lambda(w;\mathcal{B}))^p]^{1/p}$$
$$= \mathbb{E}_{x,y}[y^p(u \cdot \nabla_w \mathcal{L}_\lambda(\sigma(w \cdot x))^p]^{1/p}.$$

Applying Lemma 23 with $X = (u \cdot \nabla_w \mathcal{L}_\lambda(\sigma(w \cdot x)))^2$ and $Y = y^p(u \cdot \nabla_w \mathcal{L}_\lambda(\sigma(w \cdot x)))^{p-2}$ gives:

$$\mathbb{E}_{x,y}[(u \cdot v)^p]^{1/p} \lesssim \text{poly}(p)\tilde{O}\left(\frac{\min\left(1+\lambda^2,\sqrt{d}\right)^{-\frac{k^\star-1}{p}}}{\sqrt{1+\lambda^2}}\right)$$

$$\lesssim \text{poly}(p) \cdot \tilde{O}\left((1+\lambda^2)^{-\frac{1}{2}-\frac{k^\star-1}{p}}\right)$$

which completes the proof.  $\square$

**Corollary 2.** *Let $v, \epsilon$ be as in Lemma 14. Then for all $2 \leq p \leq d/C$,*

$$\mathbb{E}_{x,y}[\|v\|^{2p}]^{1/p} \lesssim \text{poly}(p) \cdot d \cdot \tilde{O}\left((1+\lambda^2)^{-1-\frac{k^\star-1}{p}}\right).$$

*Proof.* By Jensen's inequality,

$$\|v\|^{2p} = \mathbb{E}\left[\left(\sum_{i=1}^d (v \cdot e_i)^2\right)^p\right] \lesssim d^{p-1}\,\mathbb{E}\left[\sum_{i=1}^d (v \cdot e_i)^{2p}\right] \lesssim d^p \max_i \mathbb{E}[(z \cdot e_i)^{2p}].$$

Taking $p$th roots and using Lemma 14 finishes the proof.  $\square$

### B.4 Analyzing the Dynamics

Throughout this section we will assume $1 \leq \lambda \leq d^{1/4}$. The proof of the dynamics is split into three stages.

In the first stage, we analyze the regime $\alpha \in [\alpha_0, \lambda d^{-1/2}]$. In this regime, the signal is dominated by the smoothing.

In the second stage, we analyze the regime $\alpha \in [\lambda d^{-1/2}, 1 - o_d(1)]$. This analysis is similar to the analysis in Ben Arous et al. [1] and could be equivalently carried out with $\lambda = 0$.

Finally in the third stage, we decay the learning rate linearly to achieve the optimal rate

$$n \gtrsim d^{\frac{k^\star}{2}} + \frac{d}{\epsilon}.$$

All three stages will use the following progress lemma:

**Lemma 15.** *Let $w \in S^{d-1}$ and let $\alpha := w \cdot w^\star$. Let $(x, y)$ be a fresh batch and define*

$$v := -\nabla_w L_\lambda(w; x; y), \quad z := v - \mathbb{E}_{x,y}[v], \quad w' = \frac{w + \lambda v}{\|w + \lambda v\|} \quad and \quad \alpha' := w' \cdot w^\star.$$

*Then if $\eta \lesssim \alpha\sqrt{1 + \lambda^2}$,*

$$\alpha' = \alpha + \eta(1 - \alpha^2)c_\lambda(\alpha) + Z + \tilde{O}\left(\frac{\eta^2 d\alpha}{(1 + \lambda^2)^{k^\star}}\right).$$

*where $\mathbb{E}_{x,y}[Z] = 0$ and for all $2 \le p \le d/C$,*

$$\mathbb{E}_{x,y}[Z^p]^{1/p} \le \tilde{O}(\text{poly}(p))\left[\eta(1 + \lambda^2)^{-\frac{1}{2} - \frac{(k^\star - 1)}{p}}\right]\left[\sqrt{1 - \alpha^2} + \frac{\eta d\alpha}{\sqrt{1 + \lambda^2}}\right].$$

*Furthermore, if $\lambda = O(1)$ the $\tilde{O}(\cdot)$ can be replaced with $O(\cdot)$.*

*Proof.* Because $v \perp w$ and $1 \ge \frac{1}{\sqrt{1+x^2}} \ge 1 - \frac{x^2}{2}$,

$$\alpha' = \frac{\alpha + \eta(v \cdot w^\star)}{\sqrt{1 + \eta^2\|v\|^2}} = \alpha + \eta(v \cdot w^\star) + r$$

where $|r| \le \frac{\eta^2}{2}\|v\|^2[\alpha + \eta|v \cdot w^\star|]$. Note that by Lemma 14, $\eta(v \cdot w^\star)$ has moments bounded by $\frac{\eta}{\lambda}\text{poly}(p) \lesssim \alpha\text{poly}(p)$. Therefore by Lemma 23 with $X = \|v\|^2$ and $Y = \alpha + \eta|v \cdot w^\star|$,

$$\mathbb{E}_{x,y}[r] \le \tilde{O}\left(\eta^2 \mathbb{E}[\|v\|^2]\alpha\right).$$

Plugging in the bound on $\mathbb{E}[\|v\|^2]$ from Corollary 2 gives

$$\mathbb{E}_{x,y}[\alpha'] = \alpha + \eta(1 - \alpha^2)c_\lambda(\alpha) + \tilde{O}\left(\eta^2 d\alpha(1 + \lambda^2)^{-k^\star}\right).$$

In addition, by Lemma 14,

$$\mathbb{E}_{x,y}\left[|\eta(v \cdot w^\star) - \mathbb{E}_{x,y}[\eta(v \cdot w^\star)]|^p\right]^{1/p} \lesssim \text{poly}(p) \cdot \eta \cdot \|P_w^\perp w^\star\| \cdot \tilde{O}\left((1 + \lambda^2)^{-\frac{1}{2} - \frac{k^\star - 1}{p}}\right)$$

$$= \text{poly}(p) \cdot \eta \cdot \sqrt{1 - \alpha^2} \cdot \tilde{O}\left((1 + \lambda^2)^{-\frac{1}{2} - \frac{k^\star - 1}{p}}\right).$$

Similarly, by Lemma 23 with $X = \|v\|^2$ and $Y = \|v\|^{2(p-1)}[\alpha + \eta|v \cdot w|]^p$, Lemma 14, and Corollary 2,

$$\mathbb{E}_{x,y}\left[|r_t - \mathbb{E}_{x,y}[r_t]|^p\right]^{1/p} \lesssim \mathbb{E}_{x,y}\left[|r_t|^p\right]^{1/p}$$

$$\lesssim \eta^2 \alpha\,\text{poly}(p)\tilde{O}\left(\left(\frac{d}{(1 + \lambda^2)^{k^\star}}\right)^{1/p}\left(\frac{d}{1 + \lambda^2}\right)^{\frac{p-1}{p}}\right)$$

$$= \text{poly}(p) \cdot \eta^2 d\alpha \cdot \tilde{O}\left((1 + \lambda^2)^{-1 - \frac{k^\star - 1}{p}}\right).$$

$\square$

We can now analyze the first stage in which $\alpha \in [d^{-1/2}, \lambda \cdot d^{-1/2}]$. This stage is dominated by the signal from the smoothing.

**Lemma 16** (Stage 1). *Assume that $\lambda \geq 1$ and $\alpha_0 \geq \frac{1}{C}d^{-1/2}$. Set*

$$\eta = \frac{d^{-\frac{k^\star}{2}}(1+\lambda^2)^{k^\star-1}}{\log(d)^C} \quad \text{and} \quad T_1 = \frac{C(1+\lambda^2)d^{\frac{k^\star-2}{2}}\log(d)}{\eta} = \tilde{O}\left(d^{k^\star-1}\lambda^{-2k^\star+4}\right)$$

*for a sufficiently large constant $C$. Then with high probability, there exists $t \leq T_1$ such that $\alpha_t \geq \lambda d^{-1/2}$.*

*Proof.* Let $\tau$ be the hitting time for $\alpha_\tau \geq \lambda d^{-1/2}$. For $t \leq T_1$, let $E_t$ be the event that

$$\alpha_t \geq \frac{1}{2}\left[\alpha_0 + \eta \sum_{j=0}^{t-1} c_\lambda(\alpha_j)\right].$$

We will prove by induction that for any $t \leq T_1$, the event: $\{E_t \text{ or } t \geq \tau\}$ happens with high probability. The base case of $t = 0$ is trivial so let $t \geq 0$ and assume the result for all $s < t$. Note that $\eta/\lambda \ll \frac{d^{-1/2}}{C} \leq \alpha_j$ so by Lemma 15 and the fact that $\lambda \geq 1$,

$$\alpha_t = \alpha_0 + \sum_{j=0}^{t-1}\left[\eta(1-\alpha_j^2)c_\lambda(\alpha_j) + Z_j + \tilde{O}\left(\eta^2 d\alpha_j \lambda^{-2k^\star}\right)\right].$$

Now note that $\mathbb{P}[E_t \text{ or } t \geq \tau] = 1 - \mathbb{P}[!E_t \text{ and } t < \tau]$ so let us condition on the event $t < \tau$. Then by the induction hypothesis, with high probability we have $\alpha_s \in [\frac{\alpha_0}{2}, \lambda d^{-1/2}]$ for all $s < t$. Plugging in the value of $\eta$ gives:

$$\eta(1-\alpha_j^2)c_\lambda(\alpha_j) + \tilde{O}\left(\eta^2 d\alpha_j \lambda^{-2k^\star}\right)$$

$$\geq \eta(1-\alpha_j^2)c_\lambda(\alpha_j) - \frac{\eta d^{-\frac{k^\star-2}{2}}\lambda^{-2}\alpha_j}{C}$$

$$\geq \frac{\eta c_\lambda(\alpha_j)}{2}.$$

Similarly, because $\sum_{j=0}^{t-1} Z_j$ is a martingale we have by Lemma 22 and Lemma 24 that with high probability,

$$\sum_{j=0}^{t-1} Z_j \lesssim \tilde{O}\left(\left[\sqrt{T_1}\cdot \eta\lambda^{-k^\star} + \eta\lambda^{-1}\right]\left[1 + \max_{j<t}\frac{\eta d\alpha_j}{\lambda}\right]\right)$$

$$\lesssim \tilde{O}\left(\sqrt{T_1}\cdot \eta\lambda^{-k^\star} + \eta\lambda^{-1}\right)$$

$$\leq \frac{d^{-1/2}}{C}.$$

where we used that $\eta d\alpha_j/\lambda \leq \eta\sqrt{d} \ll 1$. Therefore conditioned on $t \leq \tau$ we have with high probability that for all $s \leq t$:

$$\alpha_t \geq \frac{1}{2}\left[\alpha_0 + \eta \sum_{j=0}^{t-1} c_\lambda(\alpha_j)\right].$$

Now we split into two cases depending on the parity of $k^\star$. First, if $k^\star$ is odd we have that with high probability, for all $t \leq T_1$:

$$\alpha_t \gtrsim \alpha_0 + \eta t\lambda^{-1}d^{-\frac{k^\star-1}{2}} \quad \text{or} \quad t \geq \tau.$$

Now let $t = T_1$. Then we have that with high probability,

$$\alpha_t \geq \lambda d^{-1/2} \quad \text{or} \quad \tau \leq T_1$$

which implies that $\tau \leq T_1$ with high probability. Next, if $k^\star$ is even we have that with high probability

$$\alpha_t \gtrsim \alpha_0 + \frac{\eta \cdot d^{-\frac{k^\star-2}{2}}}{\lambda^2}\sum_{s=0}^{t-1}\alpha_s \quad \text{or} \quad t \geq \tau.$$

As above, by Lemma 27 the first event implies that $\alpha_{T_1} \geq \lambda d^{-1/2}$ so we must have $\tau \leq T_1$ with high probability. $\qquad\square$

Next, we consider what happens when $\alpha \geq \lambda d^{-1/2}$. The analysis in this stage is similar to the online SGD analysis in [1].

**Lemma 17** (Stage 2). *Assume that $\alpha_0 \geq \lambda d^{-1/2}$. Set $\eta, T_1$ as in Lemma 16. Then with high probability, $\alpha_{T_1} \geq 1 - d^{-1/4}$.*

*Proof.* The proof is almost identical to Lemma 16. We again have from Lemma 15

$$\alpha_t \geq \alpha_0 + \sum_{j=0}^{t-1} \left[ \eta(1 - \alpha_j^2)c_\lambda(\alpha_j) + Z_j - \tilde{O}\left(\eta^2 d\alpha_j\lambda^{-2k^\star}\right) \right].$$

First, from martingale concentration we have that

$$\sum_{j=0}^{t-1} Z_j \lesssim \tilde{O}\left( \left[ \sqrt{T_1} \cdot \eta\lambda^{-k^\star} + \eta\lambda^{-1} \right] \left[ 1 + \frac{\eta d}{\lambda} \right] \right)$$

$$\lesssim \tilde{O}\left( \left[ \sqrt{T_1} \cdot \eta\lambda^{-k^\star} + \eta\lambda^{-1} \right] \cdot \lambda \right)$$

$$\lesssim \frac{\lambda d^{-1/2}}{C}$$

where we used that $\eta \ll \frac{\lambda^2}{d}$. Therefore with high probability,

$$\alpha_t \geq \frac{\alpha_0}{2} + \sum_{j=0}^{t-1} \left[ \eta(1 - \alpha_j^2)c_\lambda(\alpha_j) - \tilde{O}\left(\eta^2 d\alpha_j\lambda^{-2k^\star}\right) \right]$$

$$\geq \frac{\alpha_0}{2} + \eta\sum_{j=0}^{t-1} \left[ (1 - \alpha_j^2)c_\lambda(\alpha_j) - \frac{\alpha_j d^{-\frac{k^\star-2}{2}}}{C\lambda^2} \right].$$

Therefore while $\alpha_t \leq 1 - \frac{1}{k^\star}$, for sufficiently large $C$ we have

$$\alpha_t \geq \frac{\alpha_0}{2} + \frac{\eta}{C^{1/2}\lambda^{k^\star}} \sum_{j=0}^{t-1} \alpha_j^{k^\star-1}.$$

Therefore by Lemma 27, we have that there exists $t \leq T_1/2$ such that $\alpha_t \geq 1 - \frac{1}{k^\star}$. Next, let $p_t = 1 - \alpha_t$. Then applying Lemma 15 to $p_t$ and using $(1 - \frac{1}{k^\star})^{k^\star} \gtrsim 1/e$ gives that if

$$r := \frac{d^{-\frac{k^\star-2}{2}}}{C\lambda^2} \quad \text{and} \quad c := \frac{1}{C^{1/2}\lambda^{k^\star}}$$

then

$$p_{t+1} \leq p_t - \eta c p_t + \eta r + Z_t$$
$$= (1 - \eta c)p_t + \eta r + Z_t.$$

Therefore,

$$p_{t+s} \leq (1 - \eta c)^s p_t + r/c + \sum_{i=0}^{s-1}(1 - \eta c)^i Z_{t+s-1-i}.$$

With high probability, the martingale term is bounded by $\lambda d^{-1/2}/C$ as before as long as $s \leq T_1$, so for $s \in [\frac{C\log(d)}{\eta c}, T_1]$ we have that $p_{t+s} \lesssim C^{-1/2}\left( \left(\frac{\lambda^2}{d}\right)^{\frac{k^\star-2}{2}} + \lambda d^{-1/2} \right) \lesssim C^{-1/2}d^{-1/4}$. Setting $s = T_1 - t$ and choosing $C$ appropriately yields $p_{T_1} \leq d^{-1/4}$, which completes the proof. $\qquad\square$

Finally, the third stage guarantees not only a hitting time but a last iterate guarantee. It also achieves the optimal sample complexity in terms of the target accuracy $\epsilon$:

**Lemma 18** (Stage 3). *Assume that $\alpha_0 \geq 1 - d^{-1/4}$. Set $\lambda = 0$ and*

$$\eta_t = \frac{C}{C^4 d + t}.$$

*for a sufficiently large constant $C$. Then for any $t \leq \exp(d^{1/C})$, we have that with high probability,*

$$\alpha_t \geq 1 - O\left(\frac{d}{d+t}\right).$$

*Proof.* Let $p_t = 1 - \alpha_t$. By Lemma 15, while $p_t \leq 1/k^\star$:

$$p_{t+1} \leq p_t - \frac{\eta_t p_t}{2C} + C\eta_t^2 d + \eta_t \sqrt{p_t} \cdot W_t + \eta_t^2 d \cdot Z_t$$

$$= \left(\frac{C^4 d + t - 2}{C^4 d + t}\right) p_t + C\eta_t^2 d + \eta_t \sqrt{p_t} \cdot W_t + \eta_t^2 d \cdot Z_t.$$

where the moments of $W_t, Z_t$ are each bounded by $\text{poly}(p)$. We will prove by induction that with probability at least $1 - t \exp(-Cd^{1/C}/e)$, we have for all $s \leq t$:

$$p_s \leq \frac{2C^3 d}{C^4 d + s} \leq \frac{2}{C} \leq \frac{1}{k^\star}.$$

The base case is clear so assume the result for all $s \leq t$. Then from the recurrence above,

$$p_{t+1} \leq p_0 \frac{C^8 d^2}{(C^4 d + t)^2} + \frac{1}{(C^4 d + t)^2} \sum_{j=0}^{t} (C^4 d + j)^2 \left[C\eta_j^2 d + \eta_t \sqrt{p_t} \cdot W_t + \eta_t^2 d \cdot Z_t\right].$$

First, because $p_0 \leq d^{-1/4}$,

$$\frac{C^8 p_0 d^2}{(C^4 d + t)^2} \leq \frac{C^4 p_0 d}{C^4 d + t} \ll \frac{d}{C^4 d + t}.$$

Next,

$$\frac{1}{(C^4 d + t)^2} \sum_{j=0}^{t} (C^4 d + j)^2 C\eta_j^2 d = \frac{1}{(C^4 d + t)^2} \sum_{j=0}^{t} C^3 d$$

$$= \frac{C^3 dt}{(C^4 d + t)^2}$$

$$\leq \frac{C^3 d}{C^4 d + t}.$$

The next error term is:

$$\frac{1}{(C^4 d + t)^2} \sum_{j=0}^{t} (C^4 d + j)^2 \eta_t \sqrt{p_t} \cdot W_t.$$

Fix $p = \frac{d^{1/C}}{e}$. Then we will bound the $p$th moment of $p_t$:

$$\mathbb{E}[p_t^p] \leq \frac{2C^3 d}{C^4 d + t} + \mathbb{E}\left[p_t^p \mathbf{1}_{p_t \geq \frac{2C^3 d}{C^4 d + s}}\right]$$

$$\leq \left(\frac{2C^3 d}{C^4 d + t}\right)^p + 2^p \, \mathbb{P}\left[p_t \geq \frac{2C^3 d}{C^4 d + t}\right]$$

$$\leq \left(\frac{2C^3 d}{C^4 d + t}\right)^p + 2^p t \exp\left(-Cd^{1/C}\right).$$

Now note that because $t \leq \exp(d^{1/C})$,

$$\log\left(t \exp\left(-Cd^{1/C}\right)\right) = \log(t) - Cd^{1/C} \leq -\log(t)(C-1)d^{1/C} \leq -p\log(t).$$

Therefore $\mathbb{E}[p_t^p]^{1/p} \leq \frac{4C^3 d}{C^4 d + t}$. Therefore the $p$ norm of the predictable quadratic variation of the next error term is bounded by:

$$\text{poly}(p) \sum_{j=0}^{t} (C^4 d + j)^4 \eta_t^2 \, \mathbb{E}[p_t^p]^{1/p} \leq \text{poly}(p) \sum_{j=0}^{t} C^5 d(C^4 d + j)$$

$$\lesssim \text{poly}(p) C^5 dt(C^4 d + t).$$

In addition, the $p$ norm of the largest term in this sum is bounded by

$$\text{poly}(p) \sqrt{C^5 d(C^4 d + t)}.$$

Therefore by Lemma 22 and Lemma 24, we have with probability at least $1 - \exp(Cd^{-1/C}/e)$, this term is bounded by

$$\frac{\sqrt{C^5 dt}}{(C^4 d + t)^{3/2}} \cdot d^{1/2} \leq \frac{C^3 d}{C^4 d + t}.$$

Finally, the last term is similarly bounded with probability at least $1 - \exp(-Cd^{-1/C}/e)$ by

$$\frac{C^2 d \sqrt{t}}{(C^4 d + t)^2} \cdot d^{1/2} \ll \frac{C^3 d}{C^4 d + t}$$

which completes the induction. $\qquad\square$

We can now combine the above lemmas to prove Theorem 1:

*Proof of Theorem 1.* By Lemmas 16 to 18, if $T = T_1 + T_2$ we have with high probability for all $T_2 \leq \exp(d^{1/C})$:

$$\alpha_T \geq 1 - O\left(\frac{d}{d + T_2}\right).$$

Next, note that by Bernoulli's inequality $((1 + x)^n \geq 1 + nx)$, we have that $1 - \alpha^k \leq k(1 - \alpha)$. Therefore,

$$\begin{aligned}
L(w_T) &= \sum_{k \geq 0} \frac{c_k^2}{k!}[1 - \alpha_T^k] \\
&\leq (1 - \alpha_T) \sum_{k \geq 0} \frac{c_k^2}{(k-1)!} \\
&= (1 - \alpha_T) \, \mathbb{E}_{x \sim N(0,1)}[\sigma'(x)^2] \\
&\lesssim \frac{d}{d + T_2}
\end{aligned}$$

which completes the proof of Theorem 1. $\qquad\square$

### B.5   Proof of Theorem 2

We directly follow the proof of Theorem 2 in Damian et al. [6] which is reproduced here for completeness. We begin with the following general CSQ lemma which can be found in Szörényi [43], Damian et al. [6]:

**Lemma 19.** *Let $\mathcal{F}$ be a class of functions and $\mathcal{D}$ be a data distribution such that*

$$\mathbb{E}_{x \sim \mathcal{D}}[f(x)^2] = 1 \quad \text{and} \quad |\mathbb{E}_{x \sim \mathcal{D}}[f(x)g(x)]| \leq \epsilon \qquad \forall f \neq g \in \mathcal{F}.$$

*Then any correlational statistical query learner requires at least $\frac{|\mathcal{F}|(\tau^2 - \epsilon)}{2}$ queries of tolerance $\tau$ to output a function in $\mathcal{F}$ with $L^2(\mathcal{D})$ loss at most $2 - 2\epsilon$.*

First, we will construct a function class from a subset of $\mathcal{F} := \{\sigma(w \cdot x) : w \in S^{d-1}\}$. By [6, Lemma 3], for any $\epsilon$ there exist $\frac{1}{2}e^{c\epsilon^2 d}$ unit vectors $w_1, \ldots, w_s$ such that their pairwise inner products are all bounded by $\epsilon$. Let $\widehat{\mathcal{F}} := \{\sigma(w_i \cdot x) : i \in [s]\}$. Then for $i \neq j$,

$$\left|\mathbb{E}_{x \sim N(0, I_d)}[\sigma(w_i \cdot x)\sigma(w_j \cdot x)]\right| = \left|\sum_{k \geq 0} \frac{c_k^2}{k!}(w_i \cdot w_j)^k\right| \leq |w_i \cdot w_j|^{k^\star} \leq \epsilon^{k^\star}.$$

Therefore by Lemma 19,

$$4q \geq e^{c\epsilon^2 d}(\tau^2 - \epsilon^{k^\star}).$$

Now set

$$\epsilon = \sqrt{\frac{\log\left(4q(cd)^{k^\star/2}\right)}{cd}}$$

which gives

$$\tau^2 \leq \frac{1 + \log^{k/2}(4q(cd)^{k^\star/2})}{(cd)^{k^\star/2}} \lesssim \frac{\log^{k^\star/2}(qd)}{d^{k^\star/2}}.$$

## C  Concentration Inequalities

**Lemma 20** (Rosenthal-Burkholder-Pinelis Inequality [44])**.** *Let $\{Y_i\}_{i=0}^n$ be a martingale with martingale difference sequence $\{X_i\}_{i=1}^n$ where $X_i = Y_i - Y_{i-1}$. Let*

$$\langle Y \rangle = \sum_{i=1}^n \mathbb{E}[\|X_i\|^2|\mathcal{F}_{i-1}]$$

*denote the predictable quadratic variation. Then there exists an absolute constant $C$ such that for all $p$,*

$$\|Y_n\|_p \leq C\left[\sqrt{p\|\langle Y \rangle\|_{p/2}} + p\left\|\max_i \|X_i\|\right\|_p\right].$$

The above inequality is found in Pinelis [44, Theorem 4.1]. It is often combined with the following simple lemma:

**Lemma 21.** *For any random variables $X_1, \ldots, X_n$,*

$$\left\|\max_i \|X_i\|\right\|_p \leq \left(\sum_{i=1}^n \|X_i\|_p^p\right)^{1/p}.$$

This has the immediate corollary:

**Lemma 22.** *Let $\{Y_i\}_{i=0}^n$ be a martingale with martingale difference sequence $\{X_i\}_{i=1}^n$ where $X_i = Y_i - Y_{i-1}$. Let $\langle Y \rangle = \sum_{i=1}^n \mathbb{E}[\|X_i\|^2|\mathcal{F}_{i-1}]$ denote the predictable quadratic variation. Then there exists an absolute constant $C$ such that for all $p$,*

$$\|Y_n\|_p \leq C\left[\sqrt{p\|\langle Y \rangle\|_{p/2}} + pn^{1/p}\max_i \|X_i\|_p\right].$$

We will often use the following corollary of Holder's inequality to bound the operator norm of a product of two random variables when one has polynomial tails:

**Lemma 23.** *Let $X, Y$ be random variables with $\|Y\|_p \leq \sigma_Y p^C$. Then,*

$$\mathbb{E}[XY] \leq \|X\|_1 \cdot \sigma_Y \cdot (2e)^C \cdot \max\left(1, \frac{1}{C}\log\left(\frac{\|X\|_2}{\|X\|_1}\right)\right)^C.$$

*Proof.* Fix $\epsilon \in [0, 1]$. Then using Holder's inequality with $1 = 1 - \epsilon + \frac{\epsilon}{2} + \frac{\epsilon}{2}$ gives:

$$\mathbb{E}[XY] = \mathbb{E}[X^{1-\epsilon} X^\epsilon Y] \leq \|X\|_1^{1-\epsilon} \|X\|_2^\epsilon \|Y\|_{2/\epsilon}.$$

Using the fact that $X, Y$ have polynomial tails we can bound this by

$$\mathbb{E}[XY] = \mathbb{E}[X^{1-\epsilon} X^\epsilon Y] \leq \|X\|_1^{1-\epsilon} \|X\|_2^\epsilon \sigma_Y (2/\epsilon)^C.$$

First, if $\|X\|_2 \geq e^C \|X\|_1$, we can set $\epsilon = \dfrac{C}{\log\left(\frac{\|X\|_2}{\|X\|_1}\right)}$ which gives

$$\mathbb{E}[XY] \leq \|X\|_1 \cdot \sigma_Y \cdot \left(\frac{2e}{C} \log\left(\frac{\|X\|_2}{\|X\|_1}\right)\right)^C.$$

Next, if $\|X\|_2 \leq e^C \|X\|_1$ we can set $\epsilon = 1$ which gives

$$\mathbb{E}[XY] \leq \|X\|_2 \|Y\|_2 \leq \|X\|_1 \sigma_Y (2e)^C$$

which completes the proof. $\qquad \square$

Finally, the following basic lemma will allow is to easily convert between $p$-norm bounds and concentration inequalities:

**Lemma 24.** *Let $\delta \geq 0$ and let $X$ be a mean zero random variable satisfying*

$$\mathbb{E}[|X|^p]^{1/p} \leq \sigma_X p^C \quad for \quad p = \frac{\log(1/\delta)}{C}$$

*for some $C$. Then with probability at least $1 - \delta$, $|X| \leq \sigma_X (ep)^C$.*

*Proof.* Let $\epsilon = \sigma_X (ep)^C$. Then,

$$
\begin{aligned}
\mathbb{P}[|X| \geq \epsilon] &= \mathbb{P}[|X|^p \geq \epsilon^p] \\
&\leq \frac{\mathbb{E}[|X|^p]}{\epsilon^p} \\
&\leq \frac{(\sigma_X)^p p^{pC}}{\epsilon^p} \\
&= e^{-Cp} \\
&= \delta.
\end{aligned}
$$

$\qquad \square$

# D  Additional Technical Lemmas

The following lemma extends Steins's lemma ($\mathbb{E}_{x \sim N(0,1)}[xg(x)] = \mathbb{E}_{x \sim N(0,1)}[g'(x)]$) to the ultraspherical distribution $\mu^{(d)}$ where $\mu^{(d)}$ is the distribution of $z_1$ when $z \sim \text{Unif}(S^{d-1})$:

**Lemma 25** (Spherical Stein's Lemma). *For any $g \in L^2(\mu^{(d)})$,*

$$\mathbb{E}_{z \sim \text{Unif}(S^{d-1})}[z_1 g(z_1)] = \frac{\mathbb{E}_{z \sim \text{Unif}(S^{d+1})}[g'(z_1)]}{d}$$

*where $z_1$ denotes the first coordinate of $z$.*

*Proof.* Recall that the density of $z_1$ is equal to

$$\frac{(1 - x^2)^{\frac{d-3}{2}}}{C(d)} \quad where \quad C(d) := \frac{\sqrt{\pi} \cdot \Gamma(\frac{d-1}{2})}{\Gamma(\frac{d}{2})}.$$

Therefore,

$$\mathbb{E}_{z \sim \text{Unif}(S^{d-1})}[z_1 g(z_1)] = \frac{1}{C(d)} \int_{-1}^1 z_1 g(z_1)(1 - z_1^2)^{\frac{d-3}{2}} dz_1.$$

Now we can integrate by parts to get

$$
\mathbb{E}_{z\sim\mathrm{Unif}(S^{d-1})}[z_1 g(z_1)] = \frac{1}{C(d)}\int_{-1}^{1}\frac{g'(z_1)(1-z^2)^{\frac{d-1}{2}}}{d-1}dz_1
$$

$$
= \frac{C(d+2)}{C(d)(d-1)}\,\mathbb{E}_{z\sim\mathrm{Unif}(S^{d+1})}[g'(z_1)]
$$

$$
= \frac{1}{d}\,\mathbb{E}_{z\sim\mathrm{Unif}(S^{d+1})}[g'(z_1)].
$$

$\square$

**Lemma 26.** *For $j \le d/4$,*

$$
\mathbb{E}_{z\sim\mathrm{Unif}(S^{d-1})}\left(\frac{z_1^k}{(1-z_1^2)^j}\right) \lesssim \mathbb{E}_{z\sim\mathrm{Unif}(S^{d-2j-1})}\left(z_1^k\right).
$$

*Proof.* Recall that the PDF of $\mu^{(d)}$ is

$$
\frac{(1-x^2)^{\frac{d-3}{2}}}{C(d)} \quad \text{where} \quad C(d) := \frac{\sqrt{\pi}\cdot\Gamma(\frac{d-1}{2})}{\Gamma(\frac{d}{2})}.
$$

Using this we have that:

$$
\mathbb{E}_{z\sim\mathrm{Unif}(S^{d-1})}\left[\frac{z_1^k}{(1-z_1^2)^j}\right] = \frac{1}{C(d)}\int_{-1}^{1}\frac{x^k}{(1-x^2)^j}(1-x^2)^{\frac{d-3}{2}}dx
$$

$$
= \frac{1}{C(d)}\int_{-1}^{1}x^k(1-x^2)^{\frac{d-2j-3}{2}}dx
$$

$$
= \frac{C(d-2j)}{C(d)}\,\mathbb{E}_{z_1\sim\mu^{(d-2j)}}[z_1^k]
$$

$$
= \frac{\Gamma(\frac{d}{2})\Gamma(\frac{d-2j-1}{2})}{\Gamma(\frac{d-1}{2})\Gamma(\frac{d-2j}{2})}\,\mathbb{E}_{z\sim\mathrm{Unif}(S^{d-2j-1})}[z_1^k]
$$

$$
\lesssim \mathbb{E}_{z\sim\mathrm{Unif}(S^{d-2j-1})}[z_1^k].
$$

$\square$

We have the following generalization of Lemma 8:

**Corollary 3.** *For any $k, j \ge 0$ with $d \ge 2j+1$ and $\alpha \ge C^{-1/4}d^{1/2}$, there exist $c(j,k), C(j,k)$ such that*

$$
\mathcal{L}_\lambda\left(\frac{\alpha^k}{(1-\alpha)^j}\right) \le C(j,k)s_k(\alpha;\lambda).
$$

*Proof.* Expanding the definition of $\mathcal{L}_\lambda$ gives:

$$
\mathcal{L}_\lambda\left(\frac{\alpha^k}{(1-\alpha)^j}\right) = \mathbb{E}_{z\sim\mathrm{Unif}(S^{d-2})}\left[\frac{\left(\frac{\alpha+\lambda z_1\sqrt{1-\alpha^2}}{\sqrt{1+\lambda^2}}\right)^k}{\left(1-\frac{\alpha+\lambda z_1\sqrt{1-\alpha^2}}{\sqrt{1+\lambda^2}}\right)^j}\right].
$$

Now let $X = \frac{\lambda\sqrt{1-\alpha^2}}{\sqrt{1+\lambda^2}}\cdot\left(1-\frac{\alpha}{\sqrt{1+\lambda^2}}\right)^{-1}$ and note that by Cauchy-Schwarz, $X \le 1$. Then,

$$
\mathcal{L}_\lambda\left(\frac{\alpha^k}{(1-\alpha)^j}\right) = \frac{1}{\left(1-\frac{\alpha}{\sqrt{1+\lambda^2}}\right)^j}\,\mathbb{E}_{z\sim\mathrm{Unif}(S^{d-2})}\left[\frac{\left(\frac{\alpha+\lambda z\sqrt{1-\alpha^2}}{\sqrt{1+\lambda^2}}\right)^k}{(1-Xz_1)^j}\right]
$$

$$
\asymp \mathbb{E}_{z\sim\mathrm{Unif}(S^{d-2})}\left[\frac{(1+Xz_1)^j\left(\frac{\alpha+\lambda z\sqrt{1-\alpha^2}}{\sqrt{1+\lambda^2}}\right)^k}{(1-X^2z_1^2)^j}\right].
$$

Now we can use the binomial theorem to expand this. Ignoring constants only depending on $j, k$:

$$\mathcal{L}_\lambda\left(\frac{\alpha^k}{(1-\alpha)^j}\right) = \frac{1}{\left(1-\frac{\alpha}{\sqrt{1+\lambda^2}}\right)^j} \mathbb{E}_{z\sim\mathrm{Unif}(S^{d-2})}\left[\frac{\left(\frac{\alpha+\lambda z_1\sqrt{1-\alpha^2}}{\sqrt{1+\lambda^2}}\right)^k}{(1-Xz_1)^j}\right]$$

$$\asymp \lambda^{-k}\sum_{i=0}^k \alpha^{k-i}\lambda^i(1-\alpha^2)^{i/2}\,\mathbb{E}_{z\sim\mathrm{Unif}(S^{d-2})}\left[\frac{(1+Xz_1)^j z_1^i}{(1-X^2 z_1^2)^j}\right]$$

$$\leq \lambda^{-k}\sum_{i=0}^k \alpha^{k-i}\lambda^i(1-\alpha^2)^{i/2}\,\mathbb{E}_{z\sim\mathrm{Unif}(S^{d-2})}\left[\frac{(1+Xz_1)^j z_1^i}{(1-z_1^2)^j}\right].$$

By Lemma 26, the $z_1$ term is bounded by $d^{-\frac{i}{2}}$ when $i$ is even and $Xd^{-\frac{i+1}{2}}$ when $i$ is odd. Therefore this expression is bounded by

$$\left(\frac{\alpha}{\lambda}\right)^k \sum_{i=0}^{\lfloor\frac{k}{2}\rfloor}\left(\frac{\lambda^2(1-\alpha^2)}{\alpha^2 d}\right)^i + \left(\frac{\alpha}{\lambda}\right)^{k-1}\sum_{i=0}^{\lfloor\frac{k-1}{2}\rfloor}\alpha^{-2i}\lambda^{2i}(1-\alpha^2)^i d^{-(i+1)}$$

$$\asymp s_k(\alpha;\lambda) + \frac{1}{d}s_{k-1}(\alpha;\lambda).$$

Now note that

$$\frac{\frac{1}{d}s_{k-1}(\alpha;\lambda)}{s_k(\alpha;\lambda)} = \begin{cases} \frac{\lambda}{d\alpha} & \alpha^2 \geq \frac{\lambda^2}{d} \\ \frac{\alpha}{\lambda} & \alpha^2 \leq \frac{\lambda^2}{d} \text{ and } k^\star \text{ is even} \\ \frac{\lambda}{d\alpha} & \alpha^2 \leq \frac{\lambda^2}{d} \text{ and } k^\star \text{ is odd} \end{cases} \leq C^{-1/4}.$$

Therefore, $s_k(\alpha;\lambda)$ is the dominant term which completes the proof. □

**Lemma 27** (Adapted from Abbe et al. [4])**.** *Let $\eta, a_0 \geq 0$ be positive constants, and let $u_t$ be a sequence satisfying*

$$u_t \geq a_0 + \eta\sum_{s=0}^{t-1} u_s^k.$$

*Then, if $\max_{0\leq s\leq t-1}\eta u_s^{k-1} \leq \frac{\log 2}{k}$, we have the lower bound*

$$u_t \geq \left(a_0^{-(k-1)} - \frac{1}{2}\eta(k-1)t\right)^{-\frac{1}{k-1}}.$$

*Proof.* Consider the auxiliary sequence $w_t = a_0 + \eta\sum_{s=0}^{t-1} w_s^k$. By induction, $u_t \geq w_t$. To lower bound $w_t$, we have that

$$\eta = \frac{w_t - w_{t-1}}{w_{t-1}^k} = \frac{w_t - w_{t-1}}{w_t^k}\cdot\frac{w_t^k}{w_{t-1}^k}$$

$$\leq \frac{w_t^k}{w_{t-1}^k}\int_{w_{t-1}}^{w_t}\frac{1}{x^k}dx$$

$$= \frac{w_t^k}{w_{t-1}^k(k-1)}\left(w_{t-1}^{-(k-1)} - w_t^{-(k-1)}\right)$$

$$\leq \frac{(1+\eta w_{t-1}^{k-1})^k}{(k-1)}\left(w_{t-1}^{-(k-1)} - w_t^{-(k-1)}\right)$$

$$\leq \frac{(1+\frac{\log 2}{k})^k}{(k-1)}\left(w_{t-1}^{-(k-1)} - w_t^{-(k-1)}\right)$$

$$\leq \frac{2}{k-1}\left(w_{t-1}^{-(k-1)} - w_t^{-(k-1)}\right).$$

Therefore

$$w_t^{-(k-1)} \leq w_{t-1}^{-(k-1)} - \frac{1}{2}\eta(k-1).$$

Altogether, we get

$$w_t^{-(k-1)} \leq a_0^{-(k-1)} - \frac{1}{2}\eta(k-1)t,$$

or

$$u_t \geq w_t \geq \left( a_0^{-(k-1)} - \frac{1}{2}\eta(k-1)t \right)^{-\frac{1}{k-1}},$$

as desired. □

## E   Additional Experimental Details

To compute the smoothed loss $L_\lambda(w; x; y)$ we used the closed form for $\mathcal{L}_\lambda(He_k(w \cdot x))$ (see Appendix B.3). Experiments were run on 8 NVIDIA A6000 GPUs. Our code is written in JAX [45] and we used Weights and Biases [46] for experiment tracking.

