# OpenReview forum: "Smoothing the Landscape Boosts the Signal for SGD: Optimal Sample Complexity for Learning Single Index Models"
_NeurIPS.cc/2023/Conference — NeurIPS 2023 oral_

### Official Review · Reviewer_xzE5 · 2023-06-27

**Soundness:** 4 excellent
**Presentation:** 4 excellent
**Contribution:** 4 excellent
**Rating:** 7
**Confidence:** 3

**Summary:**

Summary: The paper closes the gap between the previously established lower bound on learning a single index model with Gaussian inputs, giving a modified learning algorithm that matches the best possible sample complexity.


**Strengths:**

Strengths: The alteration to SGD is simple but clever, and the given arguments are quite clear.  I particularly appreciate the proof sketch showing how the choice of smoothing impacts the drift term.  Tightening the gap left by [1] is a considerable contribution.

[1] Arous, Gerard Ben, Reza Gheissari, and Aukosh Jagannath. "Online stochastic gradient descent on non-convex losses from high-dimensional inference." The Journal of Machine Learning Research 22.1 (2021): 4788-4838.


**Weaknesses:**

Weaknesses: I may be misunderstanding, but it appears the experiments are using an analytic formulation of the smoothed loss, based on the hidden knowledge that the link functions (in the experiments) are hermite polynomials.  If I understand correctly, this corresponds to an “infinite synthetic sample” setting where the expectation in the smoothed loss is calculated exactly.  This wouldn’t impact the sample complexity in terms of querying the true single index model, but it raises a computational question.  Is there any reason to suspect that the analysis would change if, say, at every timestep the smoothed loss was approximated empirically with fresh samples drawn from the appropriate subset of the sphere?


**Questions:**

N/A

---

> ### Author Rebuttal · Authors · 2023-08-08
>
> We would like to thank the reviewer for the detailed and thoughtful review. Please let us know if you have any further questions or if anything is still unclear.
>
> > experiments are using an analytic formulation of the smoothed loss, based on the hidden knowledge that the link functions (in the experiments) are hermite polynomials. If I understand correctly, this corresponds to an “infinite synthetic sample” setting where the expectation in the smoothed loss is calculated exactly. This wouldn’t impact the sample complexity in terms of querying the true single index model, but it raises a computational question. Is there any reason to suspect that the analysis would change if, say, at every timestep the smoothed loss was approximated empirically with fresh samples drawn from the appropriate subset of the sphere?
>
> There are two ways to compute the smoothed gradients. The most naive way is to approximate the expectation in $L_\lambda$ using Monte Carlo. This requires roughly $d^{k^\star/2}$ draws of z at every step; however, it does not affect the sample complexity.
>
> The more efficient method is the one that we used for the experiments in section 6. It uses the observation from Lemma 7 that the smoothed gradient is a function of only two parameters: $w \cdot x$ and $\|x\|$. Explicitly, there exists a $g$ such that $\nabla L_t = g(w \cdot x, \|x\|)$ where the closed form for $g$ follows from Lemma 7. This $g$ can be efficiently numerically computed for any activation function $\sigma$ in $O(1)$ time. Furthermore, this computation only needs to be done once after which this $g$ can be reused. When $\sigma$ is a polynomial, this $g$ has a closed form (see Appendix B.3) which we used for the experiments in section 6. We’ve extended Appendix E to include a discussion of the various methods for computing the smoothed gradients.

---

### Official Review · Reviewer_mDyK · 2023-06-28

**Soundness:** 4 excellent
**Presentation:** 4 excellent
**Contribution:** 3 good
**Rating:** 8
**Confidence:** 4

**Summary:**

This paper considers the problem of learning single index models in high dimensions, i.e., functions of a high-dimensional parameter that only depend on a one-dimensional projection of this parameter. This paper is interested in the case where the link function (the function of the one-dimensional projection) is known to the statistician, but the direction of the projection is unknown.

This paper studies the sample complexity of stochastic gradient descent (SGD). Its contribution is to show that smoothing the gradients allows to take larger stepsizes, and thus improves the sample complexity. The papers draws the connection with similar results for tensor decomposition. Finally, as previous results suggests that SGD has the effect of locally averaging the gradients, the authors present their work as a first step towards improving the sample complexity bounds for SGD.

**Strengths:**

This paper is well-written; in particular, the proof sketch does a good job at explaining the technical contributions of the paper. This paper proves rigorously the remarkable phenomenon that smoothing improves the sample complexity. This is a significant steps towards understanding the benefits of stochasticity in SGD.

**Weaknesses:**

There are a few spots where I would like some clarifications, see questions below.

**Questions:**

*Remarks*

(1) Running Algorithm 1 requires to be able to compute smoothed gradients. Is it easy to compute these? Are there closed-form formulas or do we need to use sampling over $z$? It would be nice to make this explicit as it would help to understand the significance of the results: is it a suggestion for better practical performances when learning single-index models, or is it an intermediary results towards proving improved convergence bounds for SGD?

(2) Could you detail how stochastic gradient descent is a CSQ algorithm? We can show that if we use $n$ samples, then $|\hat{q} - E_{x,y}[q(x,y)]| \leq \tau / n^{1/2}$ w.h.p., but not almost surely, right? Relatedly, you write that $\tau \approx n^{-1/2}$ is a "heuristic" (l.163), thus how rigorous is the CSQ lower bound? How rigorous are the claims of "optimality"?

(3) From the proof sketch, I did not understand what is special about $\lambda = d^{1/4}$. My intuition is that smoothing with a diverging $\lambda$ should be very close to smoothing with $\lambda = \infty$. How would an algorithm with $\lambda = \infty$ (in the first steps) perform?


*Minor remarks.*

- When referring to [1], it is written many times that $n \geq d^{k^*-1}$ samples are necessary (up to log factors), however my understanding is that this assumes $k^* \geq 2$.
- l.86: what is $r$?
- Why do you consider the correlation loss rather than the square loss? I think the two would be equivalent; however it is quite unusual to optimize the correlation loss.
- Thm 1: maybe it would be good to remind that $w_0 \cdot w^* \geq d^{-1/2}$ can be guaranteed with probability 1/2.
- The statement of Thm 1 is a bit confusing. Certainly it is false that all stepsizes satisfying the O(.) bounds do not satisfy the statement (I can take the stepsizes to be 0), but maybe what you mean is that there exists a choice of stepsizes with these bounds that satisfy the statement?
- I did not understand how you count the sample complexity in the parallel with the CSQ lower bound. My understanding is that to get one query with precision $n^{-1/2}$, you need $n$ samples. So the total sample complexity should depend on the number $q$ of queries? In light of this, I did not understand l.163-164.
- Eq. below l.259: what does this tensor notation mean?


*Typo.*
- "nad" -> "and"

---

> ### Author Rebuttal · Authors · 2023-08-08
>
> We would like to thank the reviewer for the detailed and thoughtful review. We’ve corrected the typos you identified and we’ll try to clarify the higher level questions below. Please let us know if you have any further questions or if anything is still unclear.
>
> > Running Algorithm 1 requires to be able to compute smoothed gradients. Is it easy to compute these?
>
> There are two ways to compute the smoothed gradients. The most naive way is to approximate the expectation in $L_\lambda$ using Monte Carlo. This requires roughly $d^{k^\star/2}$ draws of z at every step; however, it does not affect the sample complexity.
>
> The more efficient method is the one that we used for the experiments in section 6. It uses the observation from Lemma 7 that the smoothed gradient is a function of only two parameters: $w \cdot x$ and $\|x\|$. Explicitly, there exists a $g$ such that $\nabla L_t = g(w \cdot x, \|x\|)$ where the closed form for $g$ follows from Lemma 7. This $g$ can be efficiently numerically computed for any activation function $\sigma$ in $O(1)$ time. Furthermore, this computation only needs to be done once after which this $g$ can be reused. When $\sigma$ is a polynomial, this $g$ has a closed form (see Appendix B.3) which we used for the experiments in section 6. We’ve extended Appendix E to include a discussion of the various methods for computing the smoothed gradients.
>
> > Could you detail how stochastic gradient descent is a CSQ algorithm? How rigorous are the claims of "optimality"?
>
> You are correct and the CSQ lower bound does not constitute a rigorous lower bound for gradient descent. We’ve included additional discussion in the CSQ section of our rebuttal.
>
> > From the proof sketch, I did not understand what is special about $\lambda = d^{1/4}$.
>
> This threshold shows up in the computation for the noise term in the SNR after smoothing. We’ve added an additional section in the proof sketch that computes this variance and derives the $\lambda = d^{1/4}$ threshold. For now, we’ve added this sketch to the “Computing the Variance” section of the general rebuttal. However, we will include it in the proof sketch section of the next revision of our paper.
>
> > I did not understand how you count the sample complexity in the parallel with the CSQ lower bound.
>
> The key is that each query can be answered with the same $n$ samples. In particular, any SQ algorithm can be implemented with probability $1-\delta$ with $n \ge \tau^{-2} \log(\text{queries}/\delta)$ samples: for every query $q(x,y)$ return $\frac{1}{n} \sum_{i=1}^n q(x_i,y_i)$. For fixed $q$ this will be of order $\sqrt{\frac{\log(1/\delta)}{n}}$ so the result follows from a union bound. As the query complexity only appears through a log and is generally assumed to be polynomially large in $d$, it is often omitted. See the CSQ section in our general rebuttal for additional discussion.
>
> > Eq. below l.259: what does this tensor notation mean?
>
> $M_n$ is defined by taking the $k$ tensor $T$ and iteratively contracting indices until you are left with a vector or a matrix. This is sometimes written using $\mathrm{Tr}$ notation, for example if $T$ is a 6-tensor then
> $$
> M_n = \mathrm{Tr}_{(3,4),(5,6)}(T)
> $$
> meaning that you contract the third/fourth indices and the fifth/sixth indices of $T$ to get a matrix. For our tensor notation, we use $T[A]$ to denote the contraction of $T$ with $A$ along the last $dim(A)$ dimensions of $T$ (Definition 5). In this case $A = I_d^{\otimes 2}$ is the tensor product of the identity matrix with itself and $M_n = T[A]$. This is the higher dimensional analogue of $M[I] = \langle M, I \rangle = \mathrm{tr}(M)$ when $M$ is a matrix.

---

> > ### Comment · Reviewer_mDyK · 2023-08-17
> >
> > I thank the authors for their rebuttal. Two points:
> >
> > **About the CSQ lower bound**. I now understand better the comparison with the CSQ model; in particular, it is a heuristic comparison. In light of this, many phrasings of the paper seem to be overclaiming. For instance, the claim of "optimality" in the title corresponds to a CSQ lower bound, but it is only heuristic. Most of the introduction also hides this important subtlety. Do I understand this correctly?
> >
> > It might be possible that this is a widespread confusion in the community. However, I would recommend to be much more explicit about this in the paper, especially for non-expert readers.
> >
> > **Tensor notation**. I'm not sure I understand. What does it mean to "contract" indices? Is there properly defined somewhere in the paper?
> >
> > Also, some of my remarks above were left unanswered and could be worth a clarification.

---

> > > ### Author Response · Authors · 2023-08-18
> > >
> > > We apologize for leaving some remarks unanswered. Please let us know if we missed any additional questions you had or if anything is not clear.
> > >
> > > ### About the CSQ Lower Bound
> > >
> > > We attempted to make explicit throughout the paper that the lower bound only applies to CSQ algorithms (e.g. lines 6-7 in the abstract and lines 31-32, 52-53 in the introduction). Regarding the claims of optimality, learning single index models falls under the class of problems in which there is a conjectured statistical-computational gap. In particular, the information theoretic lower bound for learning a single index model is $n \gtrsim d$, independent of $k^\star$. However, it is widely believed that no computationally efficient algorithm can solve the problem given only $O(d)$ samples for complicated link functions. Similar problems that exhibit a computational-statistical gap are tensor PCA, community detection, sparse PCA, and planted clique [1,2,3,4].
> > >
> > > None of these problems have rigorous lower bounds of the form: no polynomial time algorithm can solve this problem with fewer than $X$ samples. Such rigorous lower bounds are out of reach for current techniques. The strongest lower bounds for such problems generally use either the statistical query, low degree polynomial, or sum-of-squares framework to limit the class of learning algorithms so that a lower bound can be proven. However, it is widely believed that these statistical-computational gaps extend to all computationally efficient algorithms in the presence of mild label noise.
> > >
> > > We will update the introduction to add a discussion of the conjectured computational-statistical gap and to emphasize that $d^{k^\star/2}$ is the conjectured statistical threshold for this problem among computationally efficient algorithms, although we only prove the lower bound for the class of CSQ algorithms.
> > >
> > > [1] Feldman et al., 2012, Statistical Algorithms and a Lower Bound for Detecting Planted Cliques
> > >
> > > [2] Diakonikolas & Kane, 2017, Statistical query lower bounds for robust estimation of high-dimensional gaussians and gaussian mixtures
> > >
> > > [3] Goel et al., 2020, Statistical-Query Lower Bounds via Functional Gradients
> > >
> > > [4] Dudeja & Hsu, 2020, Statistical Query Lower Bounds for Tensor PCA
> > >
> > > ### Tensor Notation
> > >
> > > Our tensor notation is defined in Appendix A – explicitly, if $T$ is a $k$ tensor and $A$ is a $j$ tensor in $d$ dimensions with $j \le k$ then
> > > $$
> > > T[A] = \sum\_{i\_{k-j+1},\ldots,i\_k} T\_{i_1,\ldots,i\_k} A\_{i\_{k-j+1},\ldots,i\_k}
> > > $$
> > > This can also be interpreted as “flattening” $T$ into a $d^{k-j} \times d^j$ dimensional matrix, flattening $A$ into a $d^j$ dimensional vector, computing the matrix vector product, then “unflattening” the resulting $d^{k-j}$ dimensional vector into a $k-j$ tensor.
> > >
> > > Due to space constraints we cannot add Appendix A to the main paper but we will add a sentence in Section 7 that provides a reference to Appendix A for our tensor notation.
> > >
> > > ### Additional Remarks
> > >
> > > Due to the character limit our response overflowed into the next comment.

---

> > > > ### Author Response · Authors · 2023-08-18
> > > >
> > > > ### Additional Remarks
> > > >
> > > > > is it a suggestion for better practical performances when learning single-index models, or is it an intermediary results towards proving improved convergence bounds for SGD?
> > > >
> > > > Our original inspiration was to understand how the implicit regularization effects of SGD could lead to better sample complexity guarantees. However, there has also been considerable effort in the community to replace the implicit regularization of minibatch SGD by explicit regularization. We believe that directly smoothing the landscape may provide one such avenue.
> > > >
> > > > > How would an algorithm with $\lambda = \infty$  (in the first steps) perform?
> > > >
> > > > All $\lambda > d^{1/4}$ should perform equally well in the first few steps. In particular, the SNR should not decrease as you increase $\lambda$ (while $\alpha$ is still small). However, once $\alpha$ is large it is necessary to reduce the smoothing to obtain a good $\epsilon$ dependence (e.g. $d/\epsilon$ as in Theorem 1).
> > > >
> > > > > When referring to [1], it is written many times that $d^{k^\star-1}$ samples are necessary (up to log factors), however my understanding is that this assumes $k^\star \ge 2$.
> > > >
> > > > That is correct, the full sample complexity (including the $\epsilon$ dependence) is $d^{k^\star - 1} + d/\epsilon$. Because our paper focuses on the case $k^\star > 2$ (lines 137-138) we chose to omit the $d/\epsilon$ term which captures the convergence rate after SGD has escaped the low SNR regime and $\alpha \ge \Omega(1)$. However, as this changes the sample complexity when $k^\star = 1$, we will amend the discussion to be correct for any $k^\star$. In particular, dropping the $\epsilon$ dependency gives that online SGD succeeds when $n \gtrsim d^{\max(1,k^\star-1)}$ and smoothed online SGD succeeds when $n \gtrsim d^{\max(1,k^\star/2)}$.
> > > >
> > > > > I.86 what is $r$?
> > > >
> > > > Thank you for catching that – the cited result applies to multi-index models where $f^\star(x) = g(x \cdot u_1,\ldots,x \cdot u_r)$ and $r$ is the number of indices. For a single index model, $r = 1$ so the result should just be $n \gtrsim d^2$. We have corrected this typo.
> > > >
> > > > > Why do you consider the correlation loss rather than the square loss? I think the two would be equivalent; however it is quite unusual to optimize the correlation loss.
> > > >
> > > > Because our parameters are constrained to the sphere, there is no difference between the correlation and square losses in population. In particular,
> > > > $$
> > > > \mathbb{E}\_{x,y}\left[\frac{1}{2}(y - f_\theta(x))^2\right] = \mathbb{E}\_{x,y}\left[1 - yf_\theta(x)\right].
> > > > $$
> > > > The difference is that the noise structure is changed due to the additional $f_\theta(x)^2$  term. We believe that if you directly smooth the model, rather than smoothing the loss, our analysis should still go through. However, the correlation loss significantly simplifies the computations.
> > > >
> > > > > Thm 1: maybe it would be good to remind that $w_0 \cdot w^\star \gtrsim d^{-1/2}$ can be guaranteed with probability 1/2.
> > > >
> > > > Thank you for the suggestion. We've added this to the discussion immediately after Theorem 1.
> > > >
> > > > > The statement of Thm 1 is a bit confusing. Certainly it is false that all stepsizes satisfying the O(.) bounds do not satisfy the statement (I can take the stepsizes to be 0), but maybe what you mean is that there exists a choice of stepsizes with these bounds that satisfy the statement?
> > > >
> > > > Thank you for catching that. We've amended the theorem to read "There exists a choice of $T_1,\eta_t$ satisfying ..."

---

> > > > > ### Comment · Reviewer_mDyK · 2023-08-18
> > > > >
> > > > > Thank you for your quick follow-up on my response.
> > > > >
> > > > > **About the CSQ lower bound.** I think that the clarification that you provided here is great and definitely worth putting as early as in the introduction.
> > > > >
> > > > > **Tensor notation.** Indeed, I had missed the definition in the appendix. This is more clear now. Note that in Definition 5, the sum is missing (or you are using Einstein summation convention, but I didn't see written anywhere, and I don't think most readers are familiar with it).
> > > > > Also, there is a small typo l.433.
> > > > >
> > > > > Thank you also for the other answers.
> > > > >
> > > > > ----------------------
> > > > >
> > > > > To sum up, the authors have addressed very well my comments, suggesting appropriate updates when necessary. Thus I wish to keep my grade.

---

### Official Review · Reviewer_2zse · 2023-07-05

**Soundness:** 4 excellent
**Presentation:** 4 excellent
**Contribution:** 4 excellent
**Rating:** 8
**Confidence:** 4

**Summary:**

This paper studies the sample complexity of learning a single-index function \sigma(w*^Tx) via SGD on a smoothed correlation loss. The authors show that when k* is the first non-zero Hermite coefficient of \sigma, with optimally tuned smoothing, defined as averaging the loss over a sphere of radius \lambda centered at the iterate, the SGD will converge to error epsilon in d^{k*/2} + d/epsilon iterations. This improves over the iteration complexity d^{k* - 1}. This is a tight analysis, since it meets the CSQ lower bound.

The analysis is based on analyzing a certain signal-to-noise ratio which arises from comparing the alignment of the gradient with the ground truth direction, and the norm of the gradient. The authors show that when the smoothing increases, both of these terms decrease, but the norm decreases more.


**Strengths:**

- The paper achieves a significant result which enhances our understanding of SGD and achieves a known lower-bound.
- The paper is written very clearly in a way that highlights the main analysis techniques in the main body. It is useful that the authors first explain the vanilla SGD and then move on to the smoothed case.
- The paper explains the connection to related work and particularly Tensor PCA very well.

**Weaknesses:**

- Discussion of the analysis of E[|v|^2] which seems a very key part of the proof is lacking in the main body. If the authors do not have space for many more details, could they include at least some intuition or a simple example (perhaps in low dimension or for a simple sigma?) for why E[|v|^2] has the stated dependence on lambda? And then if would be helpful if there were some pointers to the appendix for where the real proofs of the main steps can be found.
- The paper only studies the correlation loss which is not frequently used in practice. Could the authors at least state why they use this, and if they believe their techniques would extend to the MSE loss?

**Questions:**

- The authors say in line 81 that the class of CSQ algorithms contains gradient descent. I am not certain this is true? Could the authors include a citation. Or perhaps it requires some qualifications?
- In line 86, I do not see a definition of r and p.
- In line 89 could the author explain (or at least define) what they mean by "ERM" or point to section 7.2
- Section 7.2 is somewhat confusing, because it skips from discussion general objectives to the Tensor PCA. The paragraph starting on line 287 is hard to understand (why will GD converge to the expectation over z of the gradient?
- Line  212 there is a v that should be a z?


**Limitations:**

The authors should discuss the limitations of using the correlation loss for learning single index functions.

---

> ### Author Rebuttal · Authors · 2023-08-08
>
> We would like to thank the reviewer for the detailed and thoughtful review. We’ve corrected the typos you identified and we’ll try to clarify the higher level questions below. Please let us know if you have any further questions or if anything is still unclear.
>
> > Discussion of the analysis of E[|v|^2] which seems a very key part of the proof is lacking in the main body.
>
> We originally omitted the variance calculation due to space constraints but as this is a crucial part of the proof sketch, we’ve added a brief sketch for the $\lambda$ dependence. See the “Computing the Variance” section of our rebuttal.
>
> > The paper only studies the correlation loss which is not frequently used in practice. Could the authors at least state why they use this, and if they believe their techniques would extend to the MSE loss?
>
> Because our parameters are constrained to the sphere, there is no difference between the correlation and MSE losses in population. In particular,
> $$
> \mathbb{E}\_{x,y}\left[\frac{1}{2}(y - f_\theta(x))^2\right] = \mathbb{E}\_{x,y}\left[1 - yf_\theta(x\right].
> $$
> The difference is that the noise structure is changed due to the additional $f_\theta(x)^2$ term. We believe that if you directly smooth the model, rather than smoothing the loss, our analysis should still go through. However, the correlation loss significantly simplifies the computations.
>
> > The authors say in line 81 that the class of CSQ algorithms contains gradient descent. I am not certain this is true? Could the authors include a citation. Or perhaps it requires some qualifications?
>
> The key connection is that GD with square loss only interacts with the labels $y$ through correlational queries. For example for GD with model $f_\theta$ and with square loss you have $$\nabla L(w;x,y) = (f_\theta(x) - \underbrace{y)\nabla f_\theta(x)}_{\text{query}}.$$
> The other term in the gradient only depends on the distribution of $x \sim N(0,I_d)$ which is known so it does not enter the sample complexity. See the CSQ section of the general rebuttal for additional discussion.
>
> > In line 89 could the author explain (or at least define) what they mean by "ERM" or point to section 7.2
>
> We used ERM to refer to empirical risk minimization where the goal is to directly minimize the empirical loss $L_n := \frac{1}{n} \sum_i L(w;x_i,y_i)$ using an algorithm like gradient descent or minibatch SGD. In this setting, each sample is seen multiple times, in contrast to the online setting studied in the rest of the paper.

---

> > ### Comment · Reviewer_2zse · 2023-08-13
> > **Read Author Rebuttal**
> >
> > Thank you for your reply. I appreciate the discussion of the variance term, and this would be great to include if there is space. If there is no space, perhaps the authors could include this sketch in the appendix.
> >
> > Yes, I see that GD with square loss is included in CSQ. Perhaps it was not clear in the paper that this was referring to square loss, could the authors specify that?

---

> > > ### Author Response · Authors · 2023-08-15
> > >
> > > Yes – we will update the exposition in Section 4 (Main Results) and we will add reference to a new section in the appendix where we clarify the connection between CSQ and GD with square/correlation loss.

---

### Official Review · Reviewer_68nG · 2023-07-11

**Soundness:** 3 good
**Presentation:** 3 good
**Contribution:** 3 good
**Rating:** 7
**Confidence:** 4

**Summary:**

This paper aims to fill the gap between the sample size complexity of online SGD and CSQ lower bound for learning a single index model. Inspired by implicit regularization of minibatch SGD, the authors show that online SGD on a smoothed correlation loss only needs sample size $n=\Omega(d^{k^*/2})$ to efficiently learn the feature direction. This smoothed loss helps us to avoid poor local minima of previous unsmoothed loss which reduces the number of samples for learning a single index model and also matches the CSQ lower bound. The authors also present a connection with a partial trace algorithm for tensor PCA.

**Strengths:**

Overall, I found the paper well-written and easy to follow. The proof sketch in Section 5 provides a clear relation among SNR of the feature alignment, the smoothed loss, and sample complexity. The paper presents many interesting insights into bridging between smoothed loss landscape and sample complexity. I believe this paper will further help us to understand the training dynamics of minibatch SGD for learning a single-index model in future work.

**Weaknesses:**

The main concern is the CSQ lower bound. The gradient-based algorithm for correlation loss is a correlation statistical query learner. But can gradient descent of square loss be fully described by correlation statistical query? There should be some additional term in gradient descent of square loss which is not a correlation query. It would be better to have a clarification on this problem.

**Questions:**

1. [Tan and Vershynin, 2019] also studied phase retrieval via online SGD and got a sharp bound similar to [1].
2. How about the misspecification case when the link target function is different from the activation function? Most of the analysis in Section 5 still works well in this case.
3. Footnote 1 on page 3: you should mention $T_w$ is the tangent space at $w$.
4. Line 125: typo $He_0(x)=1$.
5. Line 178: should it be $v_t\perp w_t$? In the analysis below Line 178, how do we ensure that the gradient norm $||v_t||=O(1)$?
6. Line 212, should it be $z\perp w$?
7. Equations below Lines 212 and 214, index $k$ should be changed into $k^*$.
8. In Section 6, for the experiments, how do you choose the batch size and learning rate? And in Figure 2, if we use the square loss for minibatch SGD training, do we need a larger sample size? It would be better to compare these two cases in the simulation to visualize the difference.
9. Line 433, typo
10. Maybe you should provide references for Lemmas 4 and 5 in Section A.2. Similar to the equation below Line 272, you should distinguish Hermite polynomial and Hermite tensor in the multivariate case.
11. In Lemma 7, you should explain the notion $z\sim\mathbb{S}^{d-2}$ and its relationship with $z_1$. Or use $\text{Unif}(\mathbb{S}^{d-2})$. Is $z_1$ the first entry of vector $z$? Similar issue for Lemma 25.
12. Equation below Line 537: should be $\nabla_w L_\lambda (w)$
13. Lemma 14, what is $\mathbb{E}_{\mathcal{B}}$?
14. Below Line 692, what is $m$? It should be $q$ in Theorem 2.
15. Equation below Line 726, $z$ in the integral should be $z_1$.

=====================================================================================================
- Tan, Y.S. and Vershynin, R., 2019. Online stochastic gradient descent with arbitrary initialization solves non-smooth, non-convex phase retrieval. arXiv preprint arXiv:1910.12837.

**Limitations:**

The limitations of the work are well addressed by the authors. I do not believe this work has any particular negative social impact.

---

> ### Author Rebuttal · Authors · 2023-08-08
>
> We would like to thank the reviewer for the detailed and thoughtful review. We’ve corrected the typos you identified and we’ll try to clarify the higher level questions below. Please let us know if you have any further questions or if anything is still unclear.
>
> > The main concern is the CSQ lower bound. The gradient-based algorithm for correlation loss is a correlation statistical query learner. But can gradient descent of square loss be fully described by correlation statistical query? There should be some additional term in gradient descent of square loss which is not a correlation query. It would be better to have a clarification on this problem.
>
> You are correct that there is an extra term, but this extra term does not interact with the labels $y$. As we are in the setting where the distribution over $x \sim N(0,I_d)$ is known, this extra term can be directly estimated by a CSQ learner. We’ve included additional details in the CSQ section of the rebuttal.
>
> > [Tan and Vershynin, 2019] also studied phase retrieval via online SGD and got a sharp bound similar to [1].
>
> Thank you, we will add this reference to the revision.
>
> > How about the misspecification case when the link target function is different from the activation function? Most of the analysis in Section 5 still works well in this case.
>
> Our result naturally extends to the misspecified setting but in a somewhat nontrivial way. Let the target link function be $\phi$ with information exponent $k^\star$ and the learner activation be $\sigma$ with information exponent $s^\star$. In the SNR calculation, the signal remains unchanged but the noise is equal to $d \lambda^{-2 s^\star}$. Going through the rest of the proof gives that the final sample complexity is $d^{k^\star - \max(1,s^\star/2)}$. Therefore when the information exponents of $\sigma$ and $\phi$ are equal, the analysis and final result are unchanged. However in the case when $\sigma$ has a lower information exponent, the sample complexity is strictly worse. We will include this discussion in the next revision of our paper.
>
> > In the analysis below Line 178, how do we ensure that the gradient norm $\|v_t\| = O(1)$?
>
> The derivation in this part of the proof sketch is heuristic so we do not directly track all of the error terms. In fact, $\|v_t\| = O(d)$ so this should be $\eta^3 d$. We believe explicitly tracking these error terms does not add to the proof sketch but to avoid being misleading we’ve replaced the $O(\eta_t^3)$ with $\ldots$ to represent the higher order terms which need to be carefully bounded.
> > In Section 6, for the experiments, how do you choose the batch size and learning rate?
>
> We give explicit formulas that we used to pick $\eta$ and the batch size (line 231-232) for all of the experiments.
>
> > And in Figure 2, if we use the square loss for minibatch SGD training, do we need a larger sample size? It would be better to compare these two cases in the simulation to visualize the difference.
>
> We believe that the sample complexity would remain unchanged so long as the smoothing is applied to the function $f$ rather than to the loss.
>
>  > Lemma 14, what is $\mathbb{E}\_\mathcal{B}$
>
> Our apologies, $\mathcal{B} = (x,y)$ denotes the current sample; however, you are correct that this was never defined. We’ve replaced this notation throughout the paper with $\mathbb{E}\_{x,y}$.

---

> > ### Comment · Reviewer_68nG · 2023-08-21
> > **Thanks for the rebuttal**
> >
> > Thank you very much to the authors for their detailed response. Regarding the clarification in the general rebuttal, I have confidence in the overall correctness of their proof and would like to increase my score to recommend an accept.

---

### Author Rebuttal · Authors · 2023-08-08

We would like to thank the reviewers for their detailed and thoughtful reviews. We’ve addressed the most common questions in this rebuttal section.

## On the CSQ lower bound

The connection between the CSQ framework and gradient descent with square loss is that GD only interacts with the labels $y$ through correlational queries. For example if the model is $f_\theta$, the gradient is equal to $$\nabla L(w;x,y) = (f_\theta(x) - \underbrace{y)\nabla f_\theta(x)}_{\text{query}}.$$
The other term in the gradient only depends on the distribution of $x \sim N(0,I_d)$ which is known so it does not enter the sample complexity. However, we emphasize that this connection is only heuristic as the errors in GD are random while the errors in the SQ/CSQ framework are adversarial. However, such SQ/CSQ bounds have been commonly used to argue the existence of statistical-computational gaps in various learning problems [1,2,3,4].

[1] Feldman et al., 2012, Statistical Algorithms and a Lower Bound for Detecting Planted Cliques

[2] Diakonikolas & Kane, 2017, Statistical query lower bounds for robust estimation of high-dimensional gaussians and gaussian mixtures

[3] Goel et al., 2020, Statistical-Query Lower Bounds via Functional Gradients

[4] Dudeja & Hsu, 2020, Statistical Query Lower Bounds for Tensor PCA

## Computing the Variance and why we need $\lambda \le d^{1/4}$.

We originally omitted the variance calculation due to space constraints but as this is a crucial part of the proof sketch, we’ve added a brief sketch for the $\lambda$ dependence. This is also the part of the proof that introduces the fundamental constraint $\lambda \le d^{1/4}$ which is crucial for the final sample complexity. Here is the sketch:

Recall that $L_\lambda(w;x,y) = 1-y\sigma(w \cdot x)$. Differentiating through the smoothing operator gives:
$$
	\nabla_w L_\lambda(w;x,y)
	= -y \nabla_w~ \mathcal{L}\_\lambda(\sigma(w \cdot x)) \approx \lambda^{-1} x \mathcal{L}\_\lambda(\sigma'(w \cdot x)).
$$
We have that $y = O(1)$ and $\|x\| = O(\sqrt{d})$ so it suffices to bound $\mathcal{L}\_\lambda(\sigma'(w \cdot x))$. The variance of this term is equal to:
$$
	\mathbb{E}\_{x}[\mathcal{L}\_\lambda(\sigma'(w \cdot x))^2] = \mathbb{E}\_x \left[ \mathbb{E}\_{z \sim \mu\_w} \left[\sigma'\left(\frac{w + \lambda z}{\sqrt{1 + \lambda^2}} \cdot x\right)\right]^2\right].
$$
To compute this expectation, we will create an i.i.d. copy $z'$ of $z$ and rewrite this expectation as:
$$
	\mathbb{E}\_{x}[\mathcal{L}\_\lambda(\sigma'(w \cdot x))^2] = \mathbb{E}\_x\left[\mathbb{E}\_{z,z' \sim \mu\_w}\left[\sigma'\left(\frac{w + \lambda z}{\sqrt{1 + \lambda^2}} \cdot x\right)\sigma'\left(\frac{w + \lambda z'}{\sqrt{1 + \lambda^2}} \cdot x\right)\right]\right].
$$
Now we can swap the expectations and compute the expectation with respect to $x$ first using the Hermite expansion of $\sigma$. As the first nonzero Hermite coefficient of $\sigma'$ is $k^\star - 1$, this variance is approximately equal to the correlation between $\frac{w + \lambda z}{\sqrt{1 + \lambda^2}}$ and $\frac{w + \lambda z'}{\sqrt{1 + \lambda^2}}$ raised to the $k^\star-1$ power:
$$
	\mathbb{E}\_{x}[\mathcal{L}\_\lambda(\sigma'(w \cdot x))^2] \approx
	\mathbb{E}\_{z,z' \sim \mu_w}\left[\left(
		\frac{w + \lambda z}{\sqrt{1 + \lambda^2}} \cdot \frac{w + \lambda z'}{\sqrt{1 + \lambda^2}}
	\right)^{k^\star-1}\right]
	= \mathbb{E}\_{z,z' \sim \mu\_w}\left[\left(\frac{1 + \lambda^2 z \cdot z'}{1 + \lambda^2}\right)^{k^\star-1}\right]
$$
As $z,z'$ are random unit vectors, their inner product is order $d^{-1/2}$. Therefore when $\lambda \le d^{1/4}$, the first term in the numerator is dominant while when $\lambda \ge d^{1/4}$, the second term is dominant. Combining these regimes gives that the variance is of order $\min(\lambda,d^{1/4})^{k^\star-1}$ which motivates our optimal choice of $\lambda = d^{1/4}$. Combining this with the fact that $y = O(1)$ and $\|x\| = O(\sqrt{d})$ gives that the gradient of the smoothed loss has variance $d\lambda^{-2k^\star}$ for $\lambda \le d^{1/4}$.

---

### Decision · Program_Chairs · 2023-09-21

**Decision:**

Accept (oral)

**Comment:**

This paper studies the task of learning a single index model, where the covariates follow an isotropic Gaussian distribution. The main goal of the paper is to fill the gap between the sample size complexity of online SGD and CSQ lower bound for learning this model. On the one hand, the existing literature shows that online SGD applied directly to the loss function has a sample complexity of $d^{k^*-1}$. On the other hand, the existing CSQ lower bound shows that $d^{k^*/2}$ samples are necessary to recover the true parameters of the problem. This paper closes this gap in the sample complexity, proving that online SGD applied to the smoothed loss function can recover the true solution with a sample complexity that matches the CSQ lower bound.

Five reviewers have reviewed the paper, and their overall assessment of the paper was extremely positive. I agree with this assessment and believe the paper answers a fundamental and impactful question.